# DiffuseBot: Breeding Soft Robots With Physics-Augmented Generative Diffusion Models

**Tsun-Hsuan Wang**[1,*]**, Juntian Zheng**[2,3]**, Pingchuan Ma**[1]**, Yilun Du**[1]**, Byungchul Kim**[1]**,**
**Andrew Spielberg**[1,4]**, Joshua B. Tenenbaum**[1]**, Chuang Gan**[1,3,5,†]**, Daniela Rus**[1,†]
[1]MIT, [2]Tsinghua University, [3]MIT-IBM Watson AI Lab, [4]Harvard, [5]UMass Amherst
https://diffusebot.github.io/

## Abstract

Nature evolves creatures with a high complexity of morphological and behavioral intelligence, meanwhile computational methods lag in approaching that diversity and efficacy. Co-optimization of artificial creatures' morphology and control *in silico* shows promise for applications in physical soft robotics and virtual character creation; such approaches, however, require developing new learning algorithms that can reason about function atop pure structure. In this paper, we present DiffuseBot, a physics-augmented diffusion model that generates soft robot morphologies capable of excelling in a wide spectrum of tasks. DiffuseBot bridges the gap between virtually generated content and physical utility by *(i)* augmenting the diffusion process with a physical dynamical simulation which provides a certificate of performance, and *ii)* introducing a co-design procedure that jointly optimizes physical design and control by leveraging information about physical sensitivities from differentiable simulation. We showcase a range of simulated and fabricated robots along with their capabilities.

## 1 Introduction

Designing dynamical virtual creatures or real-world cyberphysical systems requires reasoning about complex trade-offs in system geometry, components, and behavior. But, what if designing such systems could be made simpler, or even automated wholesale from high-level functional specifications? Freed to focus on higher-level tasks, engineers could explore, prototype, and iterate more quickly, focusing more on understanding the problem, and find novel, more performant designs. We present DiffuseBot , a first step toward efficient automatic robotic and virtual creature content creation, as an attempt at closing the stubborn gap between the wide diversity and capability of Nature *vis-a-vis* evolution, and the reiterative quality of modern soft robotics.

Specifically, we leverage diffusion-based algorithms as a means of efficiently and generatively co-designing soft robot morphology and control for target tasks. Compared with with previous approaches, DiffuseBot 's learning-based approach maintains evolutionary algorithms' ability to search over diverse forms while exploiting the efficient nature of gradient-based optimization. DiffuseBot is made possible by the revolutionary progress of AI-driven content generation, which is now able to synthesize convincing media such as images, audio, and animations, conditioned on human input. However, other than raw statistical modeling, these methods are typically task- and physics-oblivious, and tend to provide no fundamental reasoning about the performance of generated outputs. We provide the first method for bridging the gap between diffusion processes and the morphological design of cyberphysical systems, guided by physical simulation, enabling the computational creative design of virtual and physical creatures.

*This work was supported by the NSF EFRI Program (Grant No. 1830901), DSO grant DSOCO2107, DARPA Fellowship Grant HR00112110007, and MIT-IBM Watson AI Lab. † indicates equal advising.

37th Conference on Neural Information Processing Systems (NeurIPS 2023).

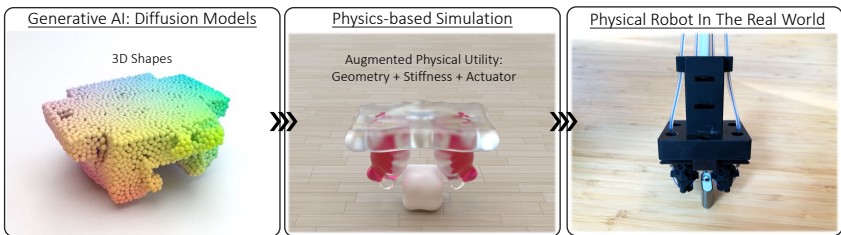

Figure 1: DiffuseBot aims to augment diffusion models with physical utility and designs for high-level functional specifications including robot geometry, material stiffness, and actuator placement.

While diffusion methods can robustly sample objects with coherent spatial structure from raw noise in a step-by-step fashion, several roadblocks preventing existing generative algorithms from being directly applied to physical soft robot co-design. First, while existing diffusion methods can generate 2D or 3D shapes, useful for, say, sampling robot geometry, they do not consider physics, nor are they directly aligned with the robotic task performance. As an alternative approach, one might consider learning a diffusion model supervised directly on a dataset of highly-performant robot designs mapped to their task performance. This leads us to the second roadblock, that is, that no such dataset exists, and, more crucially, that curating such a dataset would require a prohibitive amount of human effort and would fail to transfer to novel tasks outside that dataset.

To tackle these challenges, we propose using physical simulation to guide the generative process of pretrained large-scale 3D diffusion models. Diffusion models pretrained for 3D shapes provide an expressive base distribution that can effectively propose reasonable candidate geometries for soft robots. Next, we develop an automatic procedure to convert raw 3D geometry to a representation compatible with soft body simulation, *i.e.* one that parameterizes actuator placement and specifies material stiffness. Finally, in order to sample robots in a physics-aware and performance-driven manner, we apply two methods that leverage physically based simulation. First, we optimize the embeddings that condition the diffusion model, skewing the sampling distribution toward better-performing robots as evaluated by our simulator. Second, we reformulate the sampling process that incorporates co-optimization over structure and control. We showcase the proposed approach of DiffuseBot by demonstrating automatically synthesized, novel robot designs for a wide spectrum of tasks, including balancing, landing, crawling, hurdling, gripping, and moving objects, and demonstrate its superiority to comparable approaches. We further demonstrate DiffuseBot 's amenability to incorporating human semantic input as part of the robot generation process. Finally, we demonstrate the physical realizability of the robots generated by DiffuseBot with a proof-of-concept 3D printed real-world robot, introducing the possibility of AI-powered end-to-end CAD-CAM pipelines.

In summary, we contribute:

- A new framework that augments the diffusion-based synthesis with physical dynamical simulation in order to generatively co-design task-driven soft robots in morphology and control.
- Methods for driving robot generation in a task-driven way toward improved physical utility by optimizing input embeddings and incorporating differentiable physics into the diffusion process.
- Extensive experiments in simulation to verify the effectiveness of DiffuseBot, extensions to text-conditioned functional robot design, and a proof-of-concept physical robot as a real-world result.

## 2   Method

In this section, we first formulate the problem (Section 2.1) and then describe the proposed Diffuse-Bot framework, which consists of diffusion-based 3D shape generation (Section 2.2), a differentiable procedure that converts samples from the diffusion models into soft robots (Section 2.3), a technique to optimize embeddings conditioned by the diffusion model for improved physical utility (Section 2.4), and a reformulation of diffusion process into co-design optimization (Section 2.4).

### 2.1   Problem Formulation: Soft Robot Co-design

Soft robot co-design refers to a joint optimization of the morphology and control of soft robots. The morphology commonly involves robot geometry, body stiffness, and actuator placement. Control is the signal to the actuators prescribed by a given robot morphology. It can be formally defined as,

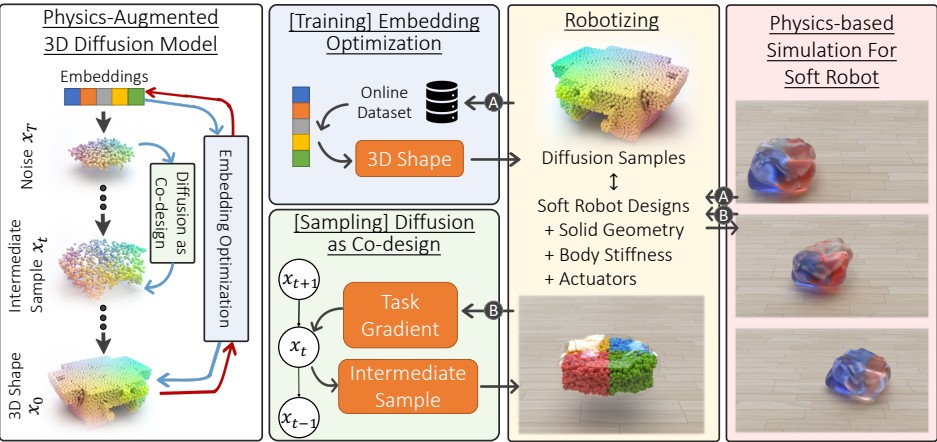

Figure 2: The DiffuseBot framework consists of three modules: (i) *robotizing*, which converts diffusion samples into physically simulatable soft robot designs (ii) *embedding optimization*, which iteratively generate new robots to be evaluated for training the conditional embedding (iii) *diffusion as co-design*, which guides the sampling process with co-design gradients from differentiable simulation. Arrow (A): evaluation of robots to guide data distribution. (B): differentiable physics as feedback.

$$\min_{\Psi,\phi} \mathcal{L}(\Psi,\phi) = \min_{\Psi,\phi} \mathcal{L}(\{\mathbf{u}_h(\mathbf{s}_h;\phi,\Psi),\mathbf{s}_h\}_{h\in[1,H]}), \quad \text{where } \mathbf{s}_{h+1} = f(\mathbf{s}_h,\mathbf{u}_h) \tag{1}$$

where $\Psi$ is robot morphology that includes geometry $\Psi_{\text{geo}}$, stiffness $\Psi_{\text{st}}$, and actuator placement $\Psi_{\text{act}}$; $\mathbf{u}_h$ is actuation with the controller's parameters $\phi$ and dependency on the robot morphology $\Psi$, $\mathbf{s}_h$ is the simulator state, $f$ is the environmental dynamics (namely the continuum mechanics of soft robots), and $H$ is robot time horizon (not to be confused with the later-on mentioned diffusion time). Co-design poses challenges in optimization including complex interdependencies between body and control variables, ambiguity between competing morphology modifications, trade-offs between flexibility and efficacy in design representations, etc. [61]. In this work, we aim at leveraging the generative power of diffusion models in searching for optimal robot designs with (1) the potential to synthesize highly-diverse robots and (2) inherent structural biases in the pre-trained 3D generative models learned from large-scale 3D datasets to achieve efficient optimization.

## 2.2 3D Shape Generation with Diffusion-based Models

Diffusion-based generative models [24, 52] aim to model a data distribution by augmenting it with auxiliary variables $\{\mathbf{x}_t\}_{t=1}^T$ defining a Gaussian diffusion process $p(\mathbf{x}_0) = \int p(\mathbf{x}_T) \prod_{t=1}^T p(\mathbf{x}_{t-1}|\mathbf{x}_t) d\mathbf{x}_{1:T}$ with the transition kernel in the forward process $q(\mathbf{x}_t|\mathbf{x}_{t-1}) = \mathcal{N}(\mathbf{x}_t; \sqrt{1-\beta_t}\mathbf{x}_{t-1}, \beta_t\mathbf{I})$ for some $0 < \beta_t < 1$. For sufficiently large $T$, we have $p(\mathbf{x}_T) \approx \mathcal{N}(\mathbf{0},\mathbf{I})$. This formulation enables an analytical marginal at any diffusion time $\mathbf{x}_t = \sqrt{\bar{\alpha}_t}\mathbf{x}_0 + \sqrt{1-\bar{\alpha}_t}\epsilon$ based on clean data $\mathbf{x}_0$, where $\epsilon \sim \mathcal{N}(\mathbf{0},\mathbf{I})$ and $\bar{\alpha}_t = \prod_{i=1}^t 1-\beta_i$. The goal of the diffusion model (or more precisely the denoiser $\epsilon_\theta$) is to learn the reverse diffusion process $p(\mathbf{x}_{t-1}|\mathbf{x}_t)$ with the loss,

$$\min_\theta \mathbb{E}_{t\sim[1,T],p(\mathbf{x}_0),\mathcal{N}(\epsilon;\mathbf{0},\mathbf{I})}[||\epsilon - \epsilon_\theta(\mathbf{x}_t(\mathbf{x}_0,\epsilon,t),t)||^2] \tag{2}$$

Intuitively, $\epsilon_\theta$ learns a one-step denoising process that can be used iteratively during sampling to convert random noise $p(\mathbf{x}_T)$ gradually into realistic data $p(\mathbf{x}_0)$. To achieve controllable generation with conditioning $\mathbf{c}$, the denoising process can be slightly altered via classifier-free guidance [25, 12],

$$\hat{\epsilon}_{\theta,\text{classifier-free}} := \epsilon_\theta(\mathbf{x}_t,t,\varnothing) + s \cdot (\epsilon_\theta(\mathbf{x}_t,t,\mathbf{c}) - \epsilon_\theta(\mathbf{x}_t,t,\varnothing)) \tag{3}$$

where $s$ is the guidance scale, $\varnothing$ is a null vector that represents non-conditioning.

## 2.3 Robotizing 3D Shapes from Diffusion Samples

We adopt Point-E [39] as a pre-trained diffusion model that is capable of generating diverse and complex 3D shapes, providing a good prior of soft robot geometries. However, the samples from the diffusion model $\mathbf{x}_t$ are in the form of surface point cloud and are not readily usable as robots to be evaluated in the physics-based simulation. Here, we describe how to robotize the diffusion samples $\mathbf{x}_t \mapsto \Psi$ and its gradient computation of the objective $\frac{d\mathcal{L}}{d\mathbf{x}_t} = \frac{\partial\mathcal{L}}{\partial\Psi_{\text{geo}}}\frac{\partial\Psi_{\text{geo}}}{\partial\mathbf{x}_t} + \frac{\partial\mathcal{L}}{\partial\Psi_{\text{st}}}\frac{\partial\Psi_{\text{st}}}{\partial\mathbf{x}_t} + \frac{\partial\mathcal{L}}{\partial\Psi_{\text{act}}}\frac{\partial\Psi_{\text{act}}}{\partial\mathbf{x}_t}$.

---

**Algorithm 1** Training: Embedding Optimization

---

**Initialize:** $\mathcal{D} \leftarrow \emptyset$, $\mathbf{c} \leftarrow \varnothing$
**while** within maximal number of epochs **do**
    Generate data with the diffusion model: $\mathbf{x}_0 \sim p_\theta(\mathbf{x}_0|\mathbf{c})$.
    Evaluate samples with physics-based simulation: $l(\mathbf{x}_0) = \mathcal{L}(\Psi(\mathbf{x}_0), \phi)$.
    Aggregate and update datasets: $\mathcal{D} \leftarrow \text{Filter}(\mathcal{D} \cup \{\mathbf{x}_0, l\})$.
    Optimize the embedding $\mathbf{c}$ on $\mathcal{D}$ using the objective (5).
**end while**

---

**Solid Geometry.** We use a Material Point Method (MPM)-based simulation [61], which takes solid geometries as inputs. This poses two obstacles: (1) conversion from surface point clouds into solid geometries, and (2) the unstructuredness of data in the intermediate samples $\mathbf{x}_t, t \neq 0$. The second issue arises from the fact that the diffusion process at intermediate steps may produce 3D points that do not form a tight surface. First, we leverage the predicted clean sample at each diffusion time $t$,

$$\hat{\mathbf{x}}_0 = \frac{\mathbf{x}_t - \sqrt{1 - \bar{\alpha}_t} \cdot \epsilon_\theta(\mathbf{x}_t, t)}{\sqrt{\bar{\alpha}_t}} \tag{4}$$

This approach is used in denoising diffusion implicit model (DDIM) [53] to approximate the unknown clean sample $\mathbf{x}_0$ in the reverse process $p(\mathbf{x}_{t-1}|\mathbf{x}_t, \hat{\mathbf{x}}_0)$. Here, we use it to construct a better-structured data for simulation. Hence, we break down the robotizing process into $\mathbf{x}_t \mapsto \hat{\mathbf{x}}_0 \mapsto \Psi$ with gradient components as $\frac{\partial \Psi}{\partial \hat{\mathbf{x}}_0} \frac{\partial \hat{\mathbf{x}}_0}{\partial \mathbf{x}_t}$, where $\frac{\partial \hat{\mathbf{x}}_0}{\partial \mathbf{x}_t}$ can be trivially derived from (4). To convert the predicted surface points $\hat{\mathbf{x}}_0$ into solid geometry, we first reconstruct a surface mesh from $\hat{\mathbf{x}}_0$, and then evenly sample a solid point cloud $\Psi_{\text{geo}}$ within its interior. For mesh reconstruction, we modify the optimization approach from Shape As Points [40], which provides a differentiable Poisson surface reconstruction that maps a control point set $\mathbf{V}_{\text{ctrl}}$ to a reconstructed surface mesh with vertices $\mathbf{V}_{\text{mesh}}$. We calculate a modified Chamfer Distance loss indicating similarity between $\mathbf{V}_{\text{mesh}}$ and $\hat{\mathbf{x}}_0$:

$$\mathcal{L}_{\text{recon}} = \frac{\lambda_{\text{mesh}}}{|\mathbf{V}_{\text{mesh}}|} \sum_{\mathbf{v} \in \mathbf{V}_{\text{mesh}}} d(\mathbf{v}, \hat{\mathbf{x}}_0) + \frac{\lambda_{\text{tagret}}}{|\hat{\mathbf{x}}_0|} \sum_{\mathbf{v} \in \hat{\mathbf{x}}_0} w(\mathbf{v}) d(\mathbf{v}, \mathbf{V}_{\text{mesh}}),$$

in which $d(\cdot, \cdot)$ denotes minimal Euclidean distance between a point and a point set, and $w(\mathbf{v})$ denotes a soft interior mask, with $w(\mathbf{v}) = 1$ for $\mathbf{v}$ outside the mesh, and $w(\mathbf{v}) = 0.1$ for $\mathbf{v}$ inside. The introduced mask term $w(v)$ aims to lower the influence of noisy points inside the mesh, which is caused by imperfect prediction of $\hat{\mathbf{x}}_0$ from noisy intermediate diffusion samples. The weight parameters are set to $\lambda_{\text{mesh}} = 1$ and $\lambda_{\text{target}} = 10$. We back-propagate $\frac{\partial \mathcal{L}_{\text{recon}}}{\partial V_{\text{mesh}}}$ to $\frac{\partial \mathcal{L}_{\text{recon}}}{\partial V_{\text{ctrl}}}$ through the differentiable Poisson solver, and then apply an Adam optimizer on $V_{\text{ctrl}}$ to optimize the loss $\mathcal{L}_{\text{recon}}$. After mesh reconstruction, the solid geometry $\Psi_{\text{geo}}$ represented by a solid interior point cloud is then sampled evenly within the mesh, with sufficient density to support the MPM-based simulation. Finally, to integrate the solidification process into diffusion samplers, we still need its gradient $\frac{\partial \Psi_{\text{geo}}}{\hat{\mathbf{x}}_0}$. We adopt Gaussian kernels on point-wise Euclidean distances as gradients between two point clouds:

$$\frac{\partial \mathbf{u}}{\partial \mathbf{v}} = \frac{\exp(-\alpha \|\mathbf{u} - \mathbf{v}\|^2)}{\sum_{v' \in \hat{\mathbf{x}}_0} \exp(-\alpha \|\mathbf{u} - \mathbf{v}'\|^2)}, \mathbf{u} \in \Psi_{\text{geo}}, \mathbf{v} \in \hat{\mathbf{x}}_0.$$

Intuitively, under such Gaussian kernels gradients, each solid point is linearly controlled by predicted surface points near it. In practice, this backward scheme works well for kernel parameter $\alpha = 20$.

**Actuators and Stiffness.** A solid geometry does not make a robot; in order for the robot to behave, its dynamics must be defined. After sampling a solid geometry, we thus need to define material properties and actuator placement. Specifically, we embed actuators in the robot body in the form of muscle fibers that can contract or expand to create deformation; further, we define a stiffness parameterization in order to determine the relationship between deformation and restorative elastic force. We adopt constant stiffness for simplicity since it has been shown to trade off with actuation strength [61]; thus we have $\Psi_{\text{st}}(\hat{\mathbf{x}}_0) = \text{const}$ and $\frac{\partial \Psi_{\text{st}}}{\partial \hat{\mathbf{x}}_0} = 0$. Then, we propose to construct actuators based on the robot solid geometry $\Psi_{\text{act}}(\Psi_{\text{geo}}(\hat{\mathbf{x}}_0))$ via clustering; namely, we perform k-means with pre-defined number of clusters on the coordinates of 3D points from the solid geometry $\Psi_{\text{geo}}$. The gradient then becomes $\frac{\partial \Psi_{\text{act}}}{\partial \Psi_{\text{geo}}} \frac{\partial \Psi_{\text{geo}}}{\partial \hat{\mathbf{x}}_0} \frac{\partial \hat{\mathbf{x}}_0}{\partial \mathbf{x}_t}$, where $\frac{\partial \Psi_{\text{act}}}{\partial \Psi_{\text{geo}}} \approx 0$ as empirically we found the clustering is quite stable in terms of label assignment, i.e., with $\Delta\Psi_{\text{geo}}$ being small, $\Delta\Psi_{\text{act}} \to 0$. Overall, we keep only the gradient for the robot geometry as empirically it suffices.

**Algorithm 2** Sampling: Diffusion As Co-design

---

**Initialize:** initial sample $\mathbf{x}_T \sim \mathcal{N}(\mathbf{0}, \mathbf{I})$
**while** within maximal number of diffusion steps $t \geq 0$ **do**
    Perform regular per-step diffusion update: $\mathbf{x}_t \Leftarrow \mathbf{x}_{t+1}$.
    **if** perform co-design **then**
        **while** within $K$ steps **do**
            Run update in (6) and (7) to overwrite $\mathbf{x}_t$.
        **end while**
    **end if**
**end while**

---

### 2.4 Physics Augmented Diffusion Model

**Embedding Optimization.** To best leverage the diversity of the generation from a large-scale pre-trained diffusion models, we propose to (1) actively generate new data from model and maintain them in a buffer, (2) use physics-based simulation as a certificate of performance, (3) optimize the embeddings conditioned by the diffusion model under a skewed data distribution to improve robotic performance in simulation. Curating a training dataset on its own alleviates the burden of manual effort to propose performant robot designs. Optimizing the conditional embeddings instead of finetuning the diffusion model eliminates the risk of deteriorating the overall generation and saves the cost of storing model weights for each new task (especially with large models). We follow,

$$\min_{\mathbf{c}} \mathbb{E}_{t \sim [1,T], p_\theta(\mathbf{x}_0|\mathbf{c}), \mathcal{N}(\epsilon; \mathbf{0}, \mathbf{I})}[||\epsilon - \epsilon_\theta(\mathbf{x}_t(\mathbf{x}_0, \epsilon, t), t, \mathbf{c})||^2] \tag{5}$$

Note the three major distinctions from (2): (i) the optimization variable is the embedding $\mathbf{c}$ not $\theta$ (ii) the denoiser is conditioned on the embeddings $\epsilon_\theta(\ldots, \mathbf{c})$ and (iii) the data distribution is based on the diffusion model $p_\theta$ not the inaccessible real data distribution $p$ and is conditioned on the embeddings $\mathbf{c}$. This adopts an online learning scheme as the sampling distribution is dependent on the changing $\mathbf{c}$. The procedure is briefly summarized in Algorithm 1, where *Filter* is an operation to drop the oldest data when exceeding the buffer limit. In addition, during this stage, we use fixed prescribed controllers since we found empirically that a randomly initialized controller may not be sufficiently informative to drive the convergence toward reasonably good solutions; also, enabling the controller to be trainable makes the optimization prohibitively slow and extremely unstable, potentially due to the difficulty of the controller required to be universal to a diverse set of robot designs. After the embedding optimization, we perform conditional generation that synthesizes samples corresponding to robot designs with improved physical utility via classifier-free guidance as in (3).

**Diffusion as Co-design.** While the optimized embedding already allows us to generate performant robots for some target tasks, we further improve the performance of individual samples by reformulating the diffusion sampling process into a co-design optimization. As described in Section 2.3, we can convert the intermediate sample at any diffusion time $\mathbf{x}_t$ to a robot design $\Psi$, *rendering an evolving robot design throughout the diffusion process*. However, regular diffusion update [24] much less resembles any gradient-based optimization techniques, which are shown to be effective in soft robot design and control with differentiable simulation [26, 2]. Fortunately, there is a synergy between diffusion models and energy-based models [54, 15, 14], which allows a more gradient-descent-like update with Markov Chain Monte Carlo (MCMC) sampling [14]. Incorporating the soft robot co-design optimization with differentiable physics [61] into the diffusion sampling process, we have

$$\text{Design Optim.:} \quad \mathbf{x}_t^{(k)} = \mathbf{x}_t^{(k-1)} + \frac{\sigma^2}{2}\left(\epsilon_\theta(\mathbf{x}_t^{(k-1)}, t) - \kappa \nabla_{\mathbf{x}_t^{(k-1)}} \mathcal{L}(\Psi(\mathbf{x}_t^{(k-1)}), \phi_t^{k-1})\right) + \sigma^2 \epsilon \tag{6}$$

$$\text{Control Optim.:} \quad \phi_t^{(k)} = \phi_t^{(k-1)} + \gamma \nabla_{\phi_t^{(k-1)}} \mathcal{L}(\Psi(\mathbf{x}_t^{(k-1)}), \phi_t^{k-1}) \tag{7}$$

where $\epsilon \sim \mathcal{N}(\mathbf{0}, \mathbf{I})$, $\mathbf{x}_t^{(0)} = \mathbf{x}_{t-1}^{(K)}$, $\kappa$ is the ratio between two types of design gradients, $K$ is the number of MCMC sampling steps at the current diffusion time, $\gamma$ is the weight for trading off design and control optimization, and $\phi_t^{(0)}$ can be either inherited from the previous diffusion time $\phi_{t-1}^{(K)}$ or reset to the initialization $\phi_T^{(0)}$. We highlight the high resemblance to gradient-based co-optimization with $\mathbf{x}_t$ as the design variable and $\phi_t$ as the control variable. This procedure is performed once every $M$ diffusion steps (Algorithm 2), where $M$ is a hyperparameter that trade-offs "guidance" strength from physical utility and sampling efficiency. Intuitively, the entire diffusion-as-co-design process is

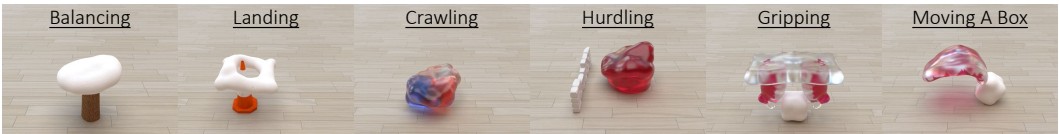

| Balancing | Landing | Crawling | Hurdling | Gripping | Moving A Box |

Figure 3: We consider passive dynamics tasks (balancing, landing), locomotion tasks (crawling, hurdling), and manipulation tasks (gripping, moving a box).

Table 1: Improved physical utility by augmenting physical simulation with diffusion models.

| Embed. Optim. | Diffusion as Co-design | Passive Dynamics | | Locomotion | | Manipulation | |
|---|---|---|---|---|---|---|---|
| | | Balancing | Landing | Crawling | Hurdling | Gripping | Moving a Box |
| | | $0.081^{.164}$ | $0.832^{.217}$ | $0.011^{.012}$ | $0.014^{.020}$ | $0.014^{.008}$ | $0.019^{.020}$ |
| ✓ | | $0.556^{.127}$ | $0.955^{.032}$ | $0.048^{.007}$ | $0.019^{.014}$ | $0.025^{.006}$ | $0.040^{.018}$ |
| ✓ | ✓ | $\mathbf{0.653}^{.107}$ | $\mathbf{0.964}^{.029}$ | $\mathbf{0.081}^{.018}$ | $\mathbf{0.035}^{.030}$ | $\mathbf{0.027}^{.004}$ | $\mathbf{0.044}^{.021}$ |

guided by three types of gradients: (i) $\epsilon_\theta(\mathbf{x}_t^{(k-1)}, \cdot)$ provides a direction for the design toward feasible 3D shapes based on the knowledge of pre-training with large-scale datasets (and toward enhanced physical utility with the optimized embeddings via classifier-free guidance using $\hat{\epsilon}_\theta(\mathbf{x}_t^{(k-1)}, \cdot, \mathbf{c})$), (ii) $\nabla_{\mathbf{x}_t}\mathcal{L}(\Psi(\mathbf{x}_t), \cdot)$ provides a direction for the design toward improving co-design objective $\mathcal{L}$ via differentiable simulation, and (iii) $\nabla_{\phi_t}\mathcal{L}(\cdot, \phi_t)$ provides a direction for the controller toward a better adaption to the current design $\mathbf{x}_t$ that allows more accurate evaluation of the robot performance.

## 3 Experiments

### 3.1 Task Setup

We cover three types of robotics tasks: passive dynamics, locomotion, and manipulation (Figure 3).

• **Passive Dynamics** tasks include balancing and landing. *Balancing* initializes the robot on a stick-like platform with small contact area with an upward velocity that introduces instability; the robot's goal is to passively balance itself after dropping on the platform. *Landing* applies an initial force to the robot toward a target; the robot's goal is to passively land as close to the target as possible.

• **Locomotion** tasks include crawling and hurdling. *Crawling* sets the robot at a rest state on the ground; the robot must actuate its body to move as far away as possible from the starting position. *Hurdling* places an obstacle in front of the robot; the robot must jump over the obstacle.

• **Manipulation** tasks include gripping and moving objects. *Gripping* places an object underneath the robot; the goal of the robot is to vertically lift the object. *Box Moving* places a box on the right end of the robot; the robot must move the box to the left.

Please refer to the appendix Section D for more detailed task descriptions and performance metrics.

### 3.2 Toward Physical Utility In Diffusion Models

**Physics-augmented diffusion.** In Table 1, we examine the effectiveness of embedding optimization and diffusion as co-design for improving physical utility. For each entry, we draw 100 samples with preset random seeds to provide valid sample-level comparison (i.e., setting the step size of co-design optimization to zero in the third row will produce almost identical samples as the second row). We report the average performance with standard deviation in the superscript. First, we observe increasing performance across all tasks while incorporating the two proposed techniques, demonstrating the efficacy of DiffuseBot. Besides, the sample-level performance does not always monotonically improve, possibly due to the stochasticity within the diffusion process and the low quality of gradient from differentiable simulation in some scenarios. For example, in gripping, when the robot fails to pick up the object in the first place, the gradient may be informative and fails to bring proper guidance toward better task performance; similarly in moving a box. In addition, we found it necessary to include control optimization during the diffusion sampling process, since, at diffusion steps further from zero, the predicted clean sample $\hat{\mathbf{x}}_0$ (derived from the intermediate sample $\mathbf{x}_t$) may differ significantly from the clean sample $\mathbf{x}_0$, leaving the prescribed controller largely unaligned.

**Comparison with baselines.** In Table 2, we compare with extensive baselines of soft robot design representation: particle-based method has each particle possessing its own distinct parameterization of design (geometry, stiffness, actuator); similarly, voxel-based method specifies design in voxel level; implicit function [37] uses use a shared multi-layer perceptron to map coordinates to design;

Table 2: Comparison with baselines.

| Methods | Passive Dynamics | | Locomotion | | Manipulation | |
|---|---|---|---|---|---|---|
| | Balancing | Landing | Crawling | Hurdling | Gripping | Moving a Box |
| Particle-based | $0.040^{.000}$ | $0.863^{.005}$ | $0.019^{.001}$ | $0.006^{.001}$ | $-0.010^{.001}$ | $0.043^{.027}$ |
| Voxel-based | $0.040^{.000}$ | $0.853^{.002}$ | $0.024^{.000}$ | $0.027^{.000}$ | $-0.009^{.000}$ | $0.025^{.022}$ |
| Implicit Function [37] | $0.106^{.147}$ | $0.893^{.033}$ | $0.043^{.024}$ | $\mathbf{0.044}^{.063}$ | $0.006^{.012}$ | $0.033^{.030}$ |
| Diff-CPPN [18] | $0.091^{.088}$ | $0.577^{.425}$ | $0.055^{.023}$ | $0.019^{.029}$ | $0.007^{.008}$ | $0.022^{.017}$ |
| DiffAqua [35] | $0.014^{.023}$ | $0.293^{.459}$ | $0.027^{.015}$ | $0.022^{.011}$ | $0.010^{.001}$ | $0.007^{.008}$ |
| DiffuseBot | $\mathbf{0.706}^{.078}$ | $\mathbf{0.965}^{.026}$ | $\mathbf{0.092}^{.016}$ | $0.031^{.011}$ | $\mathbf{0.026}^{.002}$ | $\mathbf{0.047}^{.019}$ |

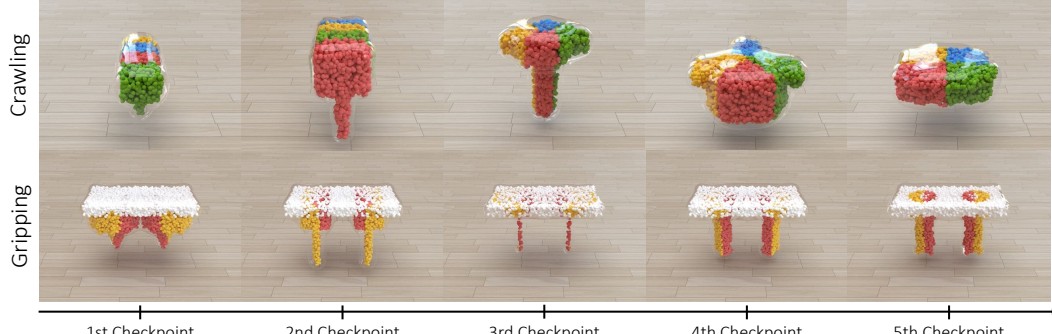

Figure 4: Examples of DiffuseBot evolving robots to solve different tasks.

DiffCPPN [18] uses a graphical model composed of a set of activation function that takes in coordinates and outputs design specification; DiffAqua [35] computes the Wasserstein barycenter of a set of aquatic creatures' meshes and we adapt to use more reasonable primitives that include bunny, car, cat, cow, avocado, dog, horse, and sofa. These baselines are commonly used in gradient-based soft robot co-design [26, 56, 61]. For each baseline method, we run the co-optimization routine for the same number of steps as in the diffusion-as-co-design stage in DiffuseBot. To avoid being trapped in the local optimum, we run each baseline with 20 different random initializations and choose the best one. Since DiffuseBot is a generative method, we draw 20 samples and report the best; this is sensible an applications-driven perspective since we only need to retrieve one performant robot, within a reasonable sample budget. We observe that our method outperforms all baselines. DiffuseBot leverages the knowledge of large-scale pre-trained models that capture the "common sense" of geometry, providing a more well-structured yet flexible prior for soft robot design.

**Soft robots bred by DiffuseBot.** In Figure 5, we demonstrate the generated soft robots that excel in locomotion and manipulation tasks. We highlight the flexibility of DiffuseBot to generate highly diverse soft robot designs that accommodate various purposes in different robotics tasks. Furthermore, in Figure 4, we show how robots evolve from a feasible yet non-necessarily functional design to an improved one that intuitively matches the task objective. By manually inspecting the evolving designs, we found that the role of the embedding optimization is to drive the diverse generations toward a converged, smaller set with elements having higher chance to succeed the task; on the other hand, the role of diffusion as co-design brings relatively minor tweaks along with alignment between the control and design. Due to space limit, we refer the reader to our project page for more results.

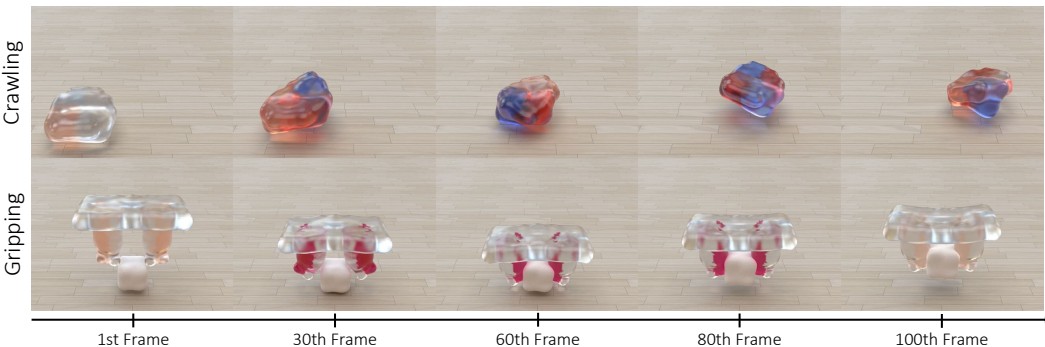

Figure 5: Examples of robots bred by DiffuseBot to achieve the desired tasks.

|  || Performance |
| --- | --- |
| MT || $0.016^{.014}$ |
| FT || $0.031^{.024}$ |
| Ours || $0.048^{.007}$ |

Table 3: Ablation on embedding optimization. MT means manually-designed text. FT means finetuning models.

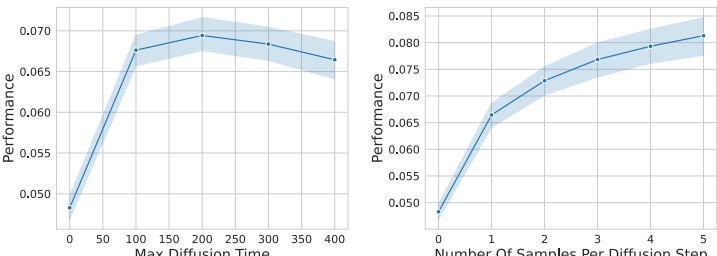

Figure 6: Varying starting point and strength of diffusion as co-design.

## 3.3 Ablation Analysis

In this section, we conduct a series of ablation studies to provide a deeper understanding of the proposed method. For simplicity, all experiments in this section are done with the crawling task.

**Embedding optimization.** In Table 3, we compare the optimization of the embedding conditioned by the diffusion models with other alternatives. The pre-trained diffusion model [39] that DiffuseBot is built upon uses CLIP embeddings [43], which allows for textual inputs. Hence, a naive approach is to manually design text for the conditional embedding of the diffusion model. The result reported in Table 3 uses *"a legged animal or object that can crawl or run fast"*. We investigated the use of text prompts; in our experience, text was difficult to optimize for *functional* robot design purposes. This is expected since most existing diffusion models perform content generation only in terms of appearance instead of physical utility, which further strengthens the purpose of this work. In addition, with exactly the same training objective as in (5), we can instead finetune the diffusion model itself. However, this does not yield better performance, as shown in the second entry in Table 3. Empirically, we found there is a higher chance of the generated samples being non-well-structured with fractured parts. This suggests that finetuning for physical utility may deteriorate the modeling of sensible 3D shapes and lead to more unstable generations.

**Diffusion as co-design.** Recall that the co-design optimization can be seamlessly incorporated into any diffusion step. In Figure 6, we examine how the strength of the injected co-design optimization affects the task performance in terms of where to apply throughout the diffusion sampling process and how many times to apply. In the left figure of Figure 6, we sweep through the maximal diffusion time of applying diffusion as co-design, i.e., for the data point at $t = 400$, we only perform co-design from $t = 400$ to $t = 0$. We found that there is a sweet spot of when to start applying co-design (at $t \approx 200$). This is because the intermediate samples at larger diffusion time $\mathbf{x}_t, t \gg 0$ are extremely under-developed, lacking sufficient connection to the final clean sample $\mathbf{x}_0$, hence failing to provide informative guidance by examining its physical utility. Furthermore, we compare against post-diffusion co-design optimization, i.e., run co-design based on the final output of the diffusion (0.064 vs ours 0.081). We allow the same computational budget by running the same number of times of differentiable simulation as in DiffuseBot. Our method performs slightly better, potentially due to the flexibility to alter the still-developing diffusion samples. Also note that while our method is interleaved into diffusion process, it is still compatible with any post-hoc computation for finetuning.

## 3.4 Flexibility To Incorporate Human Feedback

Beyond the generative power, diffusion models also provide the flexibility to composite different data distributions. This is especially useful for computational design since it empowers to easily incorporate external knowledge, e.g. from human. We follow the compositionality techniques introduced in [34, 14], which can be directly integrated into our diffusion as co-design framework. In Figure 7, we demonstrate incorporating human feedback in textual form as *"a unicorn"* into a crawling robot generated by DiffuseBot. We can see the emergence of the horn-like body part.

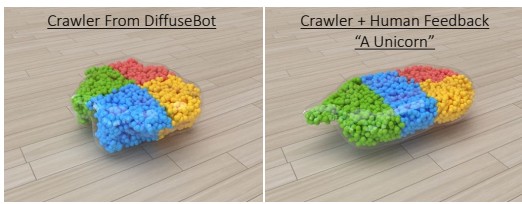

Figure 7: Incorporating human textual feedback.

## 3.5 From Virtual Generation To Physical Robot

We further fabricate a physical robot for the gripping task as a proof-of-concept to demonstrate the possibility of real-world extension. We use a 3D Carbon printer to reconstruct the exact geometry of a generated design and fill the robot body with a Voronoi lattice structure to achieve softness.

For actuators, we employ tendon transmission to realize the contraction force utilized in the soft robot gripper. In our project page, we demonstrate the robots generated by DiffuseBot are capable of picking up an object. Note that physical robot fabrication and real-world transfer have countless non-trivial challenges including stiffness and actuator design, sim-to-real gap, etc. Hence, this experiment is only meant to demonstrate the potential instead of a general, robust pipeline toward physical robots, which is left to future work. We refer the reader to the appendix for more details.

## 4 Related Work

**Heuristic Search For Soft Robot Co-Design.** Heuristic searches are simple but useful tools for co-designing soft robots. A long line of work has focused on evolutionary algorithms [7, 8, 10], with some including physical demonstrations [22, 31, 32] and recent benchmarks incorporating neural control [5]. These methods are often powered by parameterized by compositional pattern-producing networks [58], which parameterize highly expressive search spaces akin to neural networks [50, 51]. Similar to [5], [47] combines a heuristic approach with reinforcement learning, and demonstrates resulting designs on physical hardware. Other notable methods include particle-filter-based approaches [11] and simulated annealing [60]. Heuristic search methods tend to be less efficient than gradient-based or learning-based algorithms, but can reasoning about large search spaces; our approach employs the highly expressive diffusion processes, while leveraging the differentiable nature of neural networks and physical simulation for more efficient and gradient-directed search.

**Gradient-Based Soft Robot Co-Optimization.** A differentiable simulator is one in which useful analytical derivatives of any system variable with respect to any other system variable is efficiently queryable; the recent advent of soft differentiable simulation environments [26, 56, 13, 33, 41, 42, 61] has accelerated the exploration of gradient-based co-optimiation methods. [26, 56] demonstrated how differentiable simulators can be used to co-optimize very high-dimensional spatially varying material and open-loop/neural controller parameters. [38] presented gradient-based search of shape parameters for soft manipulators. Meanwhile, [61] showed how actuation and geometry could be co-optimized, while analyzing the trade-offs of design space complexity and exploration in the search procedure. DiffuseBot borrows ideas from gradient-based optimization in guiding the design search in a physics-aware way, especially in the context of control.

**Learning-Based Soft Robot Co-Design Methods.** Though relatively nascent, learning-based approaches (including DiffuseBot ) can re-use design samples to build knowledge about a problem. Further, dataset-based minibatch optimization algorithms are more robust to local minima than single-iterate pure optimization approaches. [57] demonstrated how gradient-based search could be combined with learned-models; a soft robot proprioceptive model was continually updated by simulation data from interleaved control/material co-optimization. Other work employed learning-based methods in the context of leveraging available datasets. [35] learned a parameterized representation of geometry and actuators from basis shape geometries tractable interpolation over high-dimensional search spaces. [55] leveraged motion data and sparsifying neurons to simultaneously learn sensor placement and neural soft robotic tasks such as proprioception and grasp classification.

**Diffusion Models for Content Generation.** Diffusion models [24, 52] have emerged as the de-facto standard for generating content in continuous domains such as images [44, 46], 3D content [66, 65], controls [28, 9, 1], videos [23, 49, 16], and materials [63, 62, 48]. In this paper, we explore how diffusion models in combination with differentiable physics may be used to design new robots. Most similar to our work, [64] uses differentiable physics to help guide human motion synthesis. However, while [64] uses differentiable physics to refine motions in the last few timesteps of diffusion sampling, we tightly integrate differentiable physics wih sampling throughout the diffusion sampling procedure through MCMC. We further uses differentiable simulation to define a reward objective through which we may optimize generative embeddings that represent our desirable robot structure.

## 5 Conclusion

We presented DiffuseBot, a framework that augments physics-based simulation with a diffusion process capable of generating performant soft robots for a diverse set of tasks including passive dynamics, locomotion, and manipulation. We demonstrated the efficacy of diffusion-based generation with extensive experiments, presented a method for incorporating human feedback, and prototyped a physical robot counterpart. DiffuseBot is a first step toward generative invention of soft machines, with the potential to accelerate design cycles, discover novel devices, and provide building blocks for downstream applications in automated computational creativity and computer-assisted design.

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

## A    Background On Point-E

Point-E [39] is a diffusion-based generative model that produces 3D point clouds from text or images. The Point-E pipeline consists of three stages: first, it generates a single synthetic view using a text-to-image diffusion model; second, it produces a coarse, low-resolution 3D point cloud (1024 points) using a second diffusion model which is conditioned on the generated image; third, it upsamples/"densifies" the coarse point cloud to a high-resolution one (4096 points) with a third diffusion model. The two diffusion models operating on point clouds use a permutation invariant transformer architecture with different model sizes. The entire model is trained on Point-E's curated dataset of several million 3D models and associated metadata which captures a generic distribution of common 3D shapes, providing a suitable and sufficiently diverse prior for robot geometry. The diffused data is a set of points, each point possessing 6 feature dimensions: 3 for spatial coordinates and 3 for colors. We ignore the color channels in this work. The conditioning for the synthesized image in the first stage relies on embeddings computed from a pre-trained ViT-L/14 CLIP model; in the embedding optimization of DiffuseBot, the variables to be optimized is exactly the same embedding. Diffusion as co-design is only performed in the second stage (coarse point cloud generation) since the third stage is merely an upsampling which produces only minor modifications to robot designs. We refer the reader to the original paper [39] for more details.

## B    Theoretical Motivation

**Online learning in embedding optimization.** In Section 2.4, we discuss how to online collect a dataset to optimize the embedding toward improved physical utility. Given a simplified version of (5)

$$\min_{\mathbf{c}} \mathbb{E}_{p_\theta(\mathbf{x}_0|\mathbf{c})}[g(\mathbf{x}_0, \mathbf{c})] \tag{8}$$

where, for notation simplicity, we drop $t \sim [1, T]$, $\mathcal{N}(\epsilon; \mathbf{0}, \mathbf{I})$ in the sampling distribution, and summarize $[||\epsilon - \epsilon_\theta(\mathbf{x}_t(\mathbf{x}_0, \epsilon, t), t, \mathbf{c})||^2]$ as $g(\mathbf{x}, \mathbf{c})$. We can rewrite the expectation term as,

$$\int p_\theta(\mathbf{x}_0) \frac{p_\theta(\mathbf{c}|\mathbf{x}_0)}{p_\theta(\mathbf{c})} g(\mathbf{x}_0, \mathbf{c}) dx \tag{9}$$

which allows to sample from $p_\theta(\mathbf{x}_0)$ (i.e., generating samples from the diffusion model) and reweight the loss with $\frac{p_\theta(\mathbf{c}|\mathbf{x}_0)}{p_\theta(\mathbf{c})}$; the latter scaling term is essentially proportional to a normalized task performance. Empirically, we can maintain a buffer for the online dataset and train the embedding with the sampling distribution biased toward higher task performance; we use a list to store samples with top-k performance in our implementation (we also tried reshaping the sampling distribution like prioritized experience replay in reinforcement learning but we found less stability and more hyperparameters required in training compared to our simpler top-k approach).

**Connection to MCMC.** In diffusion sampling, the simplest way to perform reverse denoising process as in Section 2.2 follows [24],

$$p_\theta(\mathbf{x}_{t-1}|\mathbf{x}_t) = \mathcal{N}\left(\frac{1}{\sqrt{\alpha_t}}(\mathbf{x}_t - \frac{\beta_t}{\sqrt{1-\bar{\alpha}_t}}\epsilon_\theta(\mathbf{x}_t, t)), \frac{1-\bar{\alpha}_{t-1}}{1-\bar{\alpha}_t}\beta_t \mathbf{I}\right) \tag{10}$$

Here, the denoising term can either be unconditional $\epsilon_\theta(\mathbf{x}_t, t)$ or conditional via classifier-free guidance $\hat{\epsilon}_{\theta,\text{classifier-free}}(\mathbf{x}_t, t, \mathbf{c})$ as in (3). We use the latter to incorporate the optimized embedding. To further leverage physics-based simulation, we aim to introduce physical utility during diffusion sampling process. One possibility is to utilize classifier-based guidance [12],

$$\hat{\epsilon}_{\theta,\text{classifier-based}} := \epsilon_\theta(\mathbf{x}_t, t) - s \cdot \sqrt{1-\bar{\alpha}_t}\nabla_{\mathbf{x}_t}\log p(\mathbf{c}|\mathbf{x}_t) \tag{11}$$

where $p(\mathbf{c}|\mathbf{x}_t)$ can be conceptually viewed as improving physical utility and $\nabla_{\mathbf{x}_t}\log p(\mathbf{c}|\mathbf{x}_t)$ can be obtained using differentiable physics and the unconditional score. Note that we slightly abuse the notation here by overloading $\mathbf{c}$ with conditioning from differentiable simulation during sampling other than the classifier-free guidance using the optimized embedding. However, combining (10) and (11) much less resembles any gradient-based optimization techniques, which are shown to be effective in soft robot co-design with differentiable simulation [26, 2]. Fortunately, drawing a connection to energy-based models [54, 15, 14], yet another alternative to incorporate conditioning in diffusion

Table 4: Configuration of embedding optimization.

|  | Balancing | Landing | Crawling | Hurdling | Gripping | Moving a Box |
|---|---|---|---|---|---|---|
| Buffer Size | 600 | 600 | 60 | 600 | 60 | 60 |
| Min. Buffer Size | 60 | 60 | 60 | 60 | 60 | 60 |
| Num. Samples / Epoch | 60 | 60 | 60 | 60 | 60 | 60 |
| Train Iter. / Epoch | 1 | 1 | 1 | 1 | 1 | 1 |
| Buffer Top-K | 12 | 12 | 6 | 6 | 6 | 6 |
| Batch Size | 6 | 6 | 6 | 6 | 6 | 6 |

Table 5: Configuration of diffusion as co-design.

|  | Balancing | Landing | Crawling | Hurdling | Gripping | Moving a Box |
|---|---|---|---|---|---|---|
| $t_{\max}$ | 400 | 150 | 400 | 400 | 400 | 400 |
| $t_{\min}$ | 0 | 0 | 0 | 0 | 0 | 0 |
| $\Delta t$ | 50 | 25 | 50 | 50 | 50 | 50 |
| $K$ | 3 | 3 | 5 | 5 | 5 | 5 |
| $\sigma$ | $10^{-4} \cdot \beta$ | $10^{-4} \cdot \beta$ | $10^{-4} \cdot \beta$ | $10^{-4} \cdot \beta$ | $10^{-4} \cdot \beta$ | $10^{-4} \cdot \beta$ |
| $\kappa$ | $10^4$ | $10^4$ | $10^4$ | $10^4$ | $10^4$ | $10^4$ |
| $\gamma$ | - | - | 0.01 | 0.001 | 0.001 | 0.001 |
| Renorm Scale | 10 | 10 | 10 | 10 | 10 | 10 |

models is Markov Chain Monte Carlo (MCMC) sampling [14], where we use Unadjusted Langevin Dynamics,

$$\mathbf{x}_t = \mathbf{x}_t^{(K)}, \quad \text{where } \mathbf{x}_t^{(k)} \sim \mathcal{N}\left(\mathbf{x}_t^{(k)}; \mathbf{x}_t^{(k-1)} + \frac{\sigma^2}{2}\nabla_{\mathbf{x}} \log p(\mathbf{c}|\mathbf{x}_t^{(k-1)}), \sigma^2 \mathbf{I}\right) \tag{12}$$

where $K$ is the number of samples in the current MCMC with $k$ as indexing, $\sigma^2$ is a pre-defined variance, and $\mathbf{x}_t^{(0)} = \mathbf{x}_{t-1}$. In the context of diffusion models, this procedure is commonly performed within a single diffusion step to drive the sample toward higher-density regime under the intermediate distribution $p(\mathbf{x}_t) = \int q(\mathbf{x}_t|\mathbf{x}_0)p(\mathbf{x}_0)d\mathbf{x}_0$ at diffusion time $t$. Inspired by its resemblance to gradient ascent with stochasticity from the added Gaussian noise of variance $\sigma^2$, we establish a connection to design optimization, reformulating diffusion process as co-design optimization as in (6) and (7). Specifically, we can apply Bayes rule to decompose the score of $p(\mathbf{c}|\mathbf{x}_t^{(k-1)})$,

$$\nabla_{\mathbf{x}} \log p(\mathbf{c}|\mathbf{x}_t) = \nabla_{\mathbf{x}} \log p(\mathbf{x}_t|\mathbf{c}) - \nabla_{\mathbf{x}} \log p(\mathbf{x}_t) \tag{13}$$

where $\nabla_{\mathbf{x}} \log p(\mathbf{x}_t)$ is simply the denoiser output $\epsilon_\theta(\mathbf{x}_t, t)$ and $\nabla_{\mathbf{x}} \log p(\mathbf{x}_t|\mathbf{c})$ is the gradient of task performance with respect to the intermediate sample of the diffusion model from differentiable physical simulation and robotizing process. Overall, this leads to (6).

## C  Implementation Details In Algorithm

In this section, we provide more implementation details and experimental configurations of Diffuse-Bot and other baselines. In Table 4, we list the configurations of the embedding optimization. With respect to Algorithm 1, "buffer" refers to the online dataset $\mathcal{D}$.

- *Buffer Size* is the capacity of the dataset.

- *Min. Buffer Size* is the minimum of data filled in the buffer before training starts.

- *Num. Samples / Epoch* is number of new samples collected in each epoch, where epoch here refers to a new round of data collection in online learning.

- *Train Iter. / Epoch* is number of training iterations per epoch.

- *Buffer Top-K* is the number of datapoints with top-k performance being retained in the *Filter* step atop the most up-to-date data.

- *Batch Size* is the batch size in the embedding optimization.

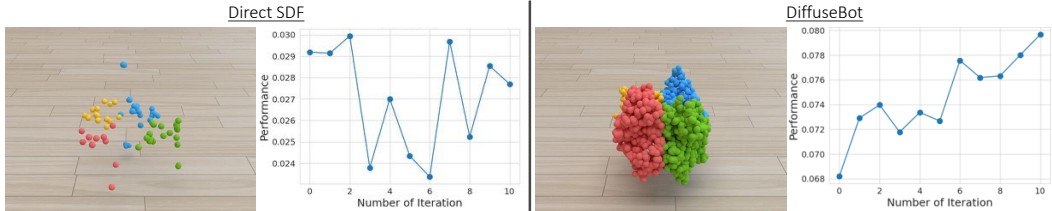

Figure 8: Comparison between *Direct SDF* and DiffuseBot on conversion to solid geometry for robotizing.

In Table 5, we list the configurations of diffusion as co-design. We follow (6)(7) for:

- $K$ is number of MCMC sampling steps at the current diffusion time.
- $\sigma$ is the standard deviation related to the MCMC step size.
- $\kappa$ is the ratio between two types of design gradients.
- $\gamma$ is the weight for trading off design and control optimization.
- $t_{\max}$ and $t_{\min}$ are the maximal and minimal diffusion time to perform diffusion as co-design, respectively.
- $\Delta t$ is the diffusion time interval to perform diffusion as co-design.

For baselines, we use learning rates for control optimization following $\gamma$ in Table 5; for particle-based and voxel-based approaches, we use learning rate 0.01 for design optimization; for implicit function and diff-CPPN, we use learning rate 0.001 for design optimization. For the inputs of implicit function and diff-CPPN, we use x, y, z coordinates in the local workspace, the distance to the workspace center on the xy, xz, yz planes, and the radius from the center. For the network architecture of implicit function, we use a 2-layer multilayer perceptron with hidden size 32 and Tanh activation. For the network architecture of diff-CPPN, we use Sin, Sigmoid, Tanh, Gaussian, SELU, Softplus, Clamped activations with 5 hidden layers and 28 graph nodes in each layer.

Hyperparameters are chosen mostly based on intuition and balancing numerical scale with very little tuning. In the following, we briefly discuss the design choices of all hyperparameters listed in Table 5 and Table 4. For min buffer size, samples per epoch, training iteration per epoch, and batch size, we roughly make sufficiently diverse the data used in the optimization and use the same setting for all tasks. For buffer size, we start with 60 and if we observe instability in optimization, we increase to 10 times, 600 (similar to online on-policy reinforcement learning); note that buffer size refers to the maximal size and increasing this won't affect runtime. For buffer Top-K, we start with 6 and if we observe limited diversity of generation throughout the optimization (or lack of exploration), we double it. For $t_{max}$, $t_{min}$, and $\Delta t$, we roughly inspect how structured the generation in terms of achieving the desired robotic task to determine $t_{max}$ and modify $\Delta t$ accordingly to match the similar number of performing MCMC sampling (e.g., $t_{max}/\Delta t$: $400/50 \approx 150/25$). For the number of MCMC steps $K$, we simply set 3 for passive tasks and 5 for active tasks by intuition. For $\sigma$, we simply follow one of the settings in [14]. For the guidance scale $\kappa$ and renorm scale, we check the numerical values between $\epsilon$ and gradient from differentiable simulation and try to make them roughly in the similar magnitude, and set the same scale for all tasks for simplicity. For $\gamma$, we set 0.001 for trajectory optimization and 0.01 for parameterized controllers based on our experience of working with differentiable physics. Overall, from our empirical findings, the only hyperparameters that may be sensitive include buffer size and buffer Top-K for optimization stability and generation diversity, and guidance scales, which need to be tuned to match the numerical magnitude of other terms so as to take proper effect.

## D Details In Task Setup

In this section, we provide more details of all tasks setup including environment configuration and prescribed actuator and controller if any (as mentioned in Section 2.4; it is fixed during embedding optimization and will be co-optimize during diffusion as co-design). We select tasks that

1. can cover a wide spectrum of existing robotics tasks: we briefly categorize tasks into passive dynamics, locomotion, and manipulation. Note that passive dynamics tasks are explicitly considered

here since there is no active control of robot bodies, making optimization on robot design a direct factor toward physical utility.

2. only involve lower-level control/motion without the complications of long-term or higher-level task planning: we select tasks that mostly involve few motor skills, e.g., in manipulation, instead of pick and place, we simply aim at picking up/gripping an object.

3. are commonly considered in other soft robot co-design literature: all proposed active tasks are widely used in the soft robot community, including crawling [8, 10, 47, 61], hurdling/jumping [26, 59, 3], and manipulating objects [5, 11, 38].

4. may induce more visible difference in robot designs between the performing and the non-performing ones to facilitate evaluation and algorithmic development: we select tasks more based on heuristics and intuition, e.g., in crawling, we expect leg-like structures may outperform other random designs.

We build our environments on top of SoftZoo [61] and employ the Material Point Method for simulation. Each environment is composed of boundary conditions that include impenetrable ground, and, in the case of fixed objects or body parts, glued particles. All tasks use units without direct real-world physical correspondence and last for 100 steps with each step consisting of 17 simulation substeps. In the following, when describing the size of a 3D shape without explicitly mentioning the axis, we follow the order: length (x, in the direction when we talk about left and right), height (y, parallel with the gravity direction), and width (z). All tasks are demonstrated in Figure 3 in the main paper.

**Balancing.** The robot is initialized atop a stick-like platform of shape 0.02-unit $\times$ 0.05-unit $\times$ 0.02-unit. The robot is given an initial upward velocity of a 0.5-unit/second and allowed to free fall under gravity. The goal is for the robot to passively balance itself after dropping again on the platform; the performance is measured as the intersection over union (IoU) between the space occupied by the robot during the first simulation step and the space occupied by the robot during the last simulation step. The robot geometry is confined to a 0.08-unit $\times$ 0.08-unit $\times$ 0.08-unit workspace. There is no prescribed actuator placement or controller (passive dynamics).

**Landing.** The robot is initialized to be 0.08-unit to the right and 0.045-unit above the landing target with size of 0.02-unit $\times$ 0.05-unit $\times$ 0.02-unit. The robot is given an initial velocity of 0.5-unit/second to the right. The goal of the robot is to land at the target; the performance is measured as the exponential to the power of the negative distance between the target and the robot in the last frame $e^{-||p_H^{\text{object}} - p_H^{\text{robot}}||}$, where $p_H$ is the position of the robot or object at the last frame with horizon $H$. The robot geometry is confined to a 0.08-unit $\times$ 0.08-unit $\times$ 0.08-unit workspace. There is no prescribed actuator placement or controller (passive dynamics).

**Crawling.** The robot is initialized at rest on the ground. The goal of the robot is to actuate its body to move as far away as possible from the starting position; the performance is measured as the distance traveled $||p_H^{x,\text{robot}} - p_0^{x,\text{robot}}||$, where $p_.^{x,\text{robot}}$ is the position of the robot in the x axis at a certain frame with horizon $H$. The robot geometry is confined to a 0.08-unit $\times$ 0.08-unit $\times$ 0.08-unit workspace. The actuator placement is computed by clustering the local coordinates of the robot centered at its average position in the xz (non-vertical) plane into 4 groups. Each group contains an actuator with its direction parallel to gravity. The prescribed controller is a composition of four sine waves with frequency as 30hz, amplitude as 0.3, and phases as $0.5\pi$, $1.5\pi$, $0$, and $\pi$ for the four actuators.

**Hurdling.** An obstacle of shape 0.01-unit $\times$ 0.03-unit $\times$ 0.12-unit is placed in 0.07-unit front of the robot. The goal of the robot is to jump as far as possible, with high distances achieved only by bounding over the obstacle. The performance is measured as the distance traveled. The robot geometry is confined to a 0.08-unit $\times$ 0.08-unit $\times$ 0.08-unit workspace. The actuator placement is computed by clustering the local coordinates of the robot centered at its average position in the length direction into 2 groups. Each group contains an actuator aligned parallel to gravity. The prescribed controller takes the form of open-loop, per-step actuation sequences, set to linearly-increasing values from (0.0, 0.0) to (1.0, 0.3) between the first and the thirtieth frames and zeros afterward for the two actuators (the first value of the aforementioned actuation corresponds to the actuator closer to the obstacle) respectively.

**Gripping.** An object of shape 0.03-unit $\times$ 0.03-unit $\times$ 0.03-unit is placed 0.08-unit underneath the robot. The goal of the robot is to vertically lift the object; the performance is measured as the vertical distance of the object being lifted $||p_H^{y,\text{object}} - p_0^{y,\text{object}}||$, where $p_.^{y,\text{object}}$ is the position of the

object in the y axis at a certain frame with horizon $H$. Within a 0.06-unit $\times$ 0.08-unit $\times$ 0.12-unit workspace, we decompose the robot into a base and two attached submodules, and we set the output of DiffuseBot or other robot design algorithms as one of the submodules and make constrain the other submodule to be a mirror copy; conceptually we design "the finger of a parallel gripper." The base is clamped/glued to the upper boundary in z and given a large polyhedral volume to serve as the attachment point of the gripper finger submodules. The actuator placement is computed by clustering the local coordinates of the robot centered at its average position in the length direction into 2 groups; each submodule has one pair of actuators. Each actuator is aligned parallel to gravity. Overall, one pair of actuators comprises the "inner" part of the gripper and the other comprises the "outer" part. Suppose the actuation of the two pairs of actuators is denoted in the format of (actuation of the outer part, actuation of the inner part), the prescribed controller is per-frame actuation, set to (i) linearly-increasing values from (0.0, 0.0) to (0.0, 1.0) between the first and the fiftieth frames, and (ii) then linearly-decreasing values from (1.0, 0.0) to (0.0, 0.0) between the fiftieth and the last frames.

**Moving a box.** A 0.03-unit $\times$ 0.03-unit $\times$ 0.03-unit cube is placed on the right end of the robot (half a body length of the robot to the right from the robot center). The goal of the robot is to move the box to the left; the performance is measured as the distance that the box is moved to the right with respect to its initial position $||p_H^{x,\text{object}} - p_0^{x,\text{object}}||$, where $p^{x,\text{object}}$ is the position of the object in the x axis at a certain frame with horizon $H$. The robot geometry is confined to a 0.16-unit $\times$ 0.06-unit $\times$ 0.06-unit workspace. The actuator placement is computed by clustering the local coordinates of the robot centered at its average position in the height direction into 2 groups. Each group contains an actuator aligned parallel to the ground. The prescribed controller is per-frame actuation, initialized to linearly-increasing values from (0.0, 0.0) to (1.0, 0.0) between the first and last frame for the lower and the upper actuator respectively.

# E    Analysis On Robotizing

As mentioned in Section 2.3, Material Point Method simulation requires solid geometry for simulation; thus, we need to convert the surface point cloud from Point-E [39] to a volume. The most direct means of converting the point cloud to a solid geometry is to compute the signed distance function (SDF), and populate the interior of the SDF using rejection sampling. We refer to this baseline as *Direct SDF*. Here, we use a pretrained transformer-based model provided by Point-E as the SDF. In Figure 8, we compare Direct SDF with our approach described in Section 2.3 Solid Geometry. We perform robotizing on intermediate samples at t=300. We observe that Direct SDF fails to produce well-structured solid geometry since it is trained with watertight on geometry, and thus the conversion cannot gracefully handle generated point clouds that do not exhibit such watergith structure. This is specifically common in the intermediate diffusion sample $\mathbf{x}_t$ as $\mathbf{x}_t$ is essentially a Gaussian-noise-corrupted version of the clean surface point cloud. In contrast, the robotizing of DiffuseBot produces a much well-structured solid geometry since it explicitly handles the noisy interior 3D points by introducing a tailored loss as in Shape As Points optimization [40] (see Section 2.3). In addition, a better-structured robot geometry is critical to obtain not only a more accurate evaluation of a robot design at the forward pass of the simulation but also the gradients from differentiable physics at the backward pass. In Figure 8, we further perform co-optimization on the two robots obtained by Direct SDF and DiffuseBot; we observe a more stably increasing trend of task performance in our approach, demonstrating that the gradient is usually more informative with a better-structured robot. Empirically, while Direct SDF sometimes still provides improving trend with unstructured robots in co-design optimization, the performance gain is not as stable as DiffuseBot.

# F    Additional Visualizations Of Experiments

In this section, we show more visualization for a diverse set of the experimental analysis. Please visit our project page (https://diffusebot.github.io/) for animated videos.

**Generated robots performing various tasks.** In Figure 9, we display a montage of a generated robot's motion for each task; this is the unabridged demonstration of Figure 5 in the main paper.

**Physics-guided robot generation** In Figure 10, we show how DiffuseBot evolves robots throughout the embedding optimization process; this is the full demonstration of Figure 4 in the main paper. Note

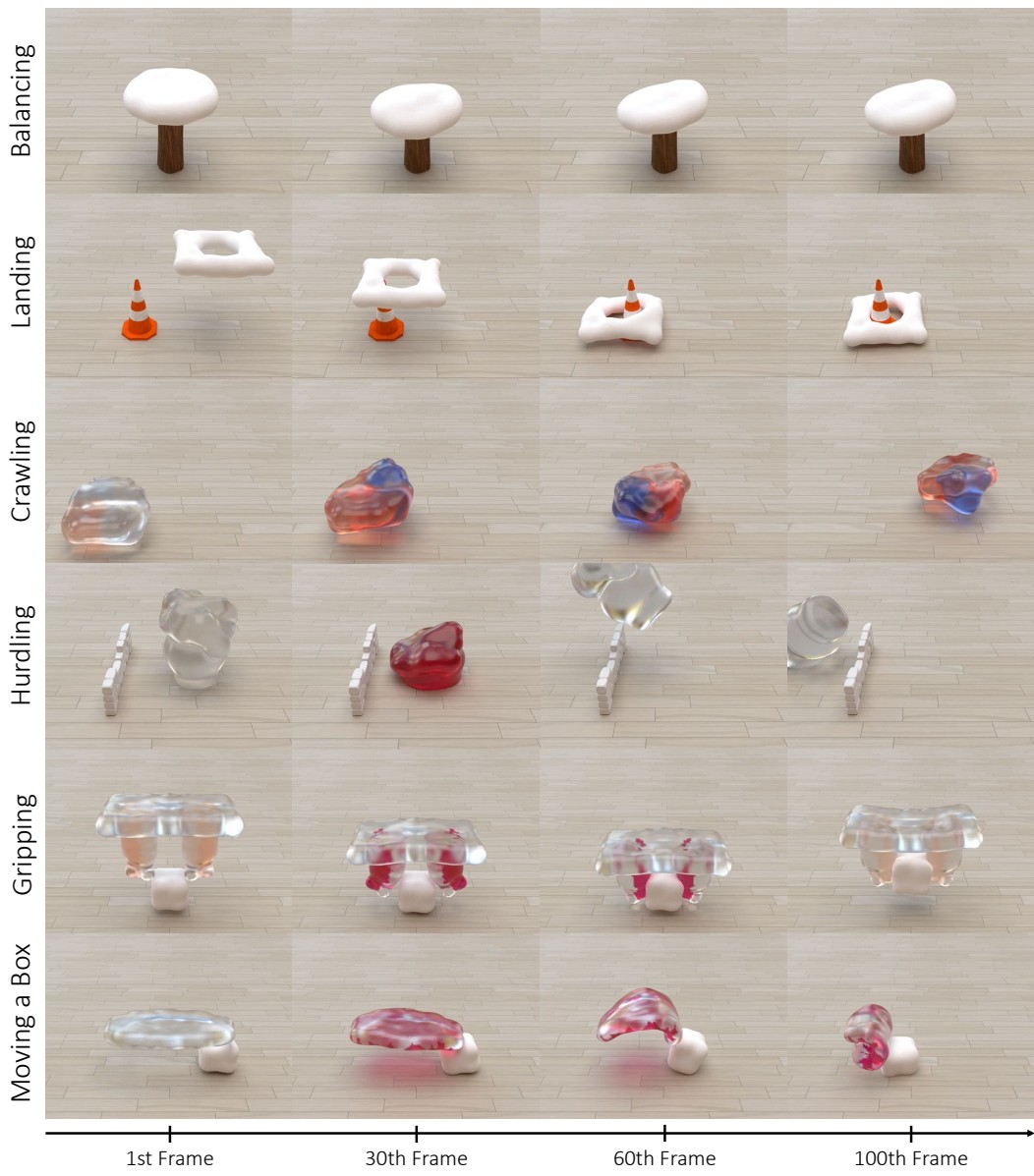

Figure 9: Examples of robots bred by DiffuseBot to achieve the all presented tasks.

that DiffuseBot is a generative algorithm and thus results presented are for demonstration purposes and reflect only a single fixed random seed.

**Diffusion process of robot generation.** In Figure 11, we show how robots evolve throughout the diffusion process at different diffusion time via $\mathbf{x}_t$ with $t$ from $T$ to $0$; not to be confused by Figure 10 where all generations are obtained *via* a full diffusion sampling $\mathbf{x}_0$ and different conditioning embeddings $\mathbf{c}$. This figure demonstrates the robotized samples across diffusion times, gradually converging to the final functioning robots as $t$ approaches $0$.

**Comparison with baselines.** In Figure 12, we present robots generated from different baselines. Note that Point-E is essentially DiffuseBot without embedding optimization and diffusion as co-design.

**Incorporating human feedback.** In Figure 13, we showcase more examples of incorporating human textual feedback to the generation process of DiffuseBot. As briefly mentioned in Section 3.4 in the main paper, we leverage the compositionality of diffusion-based generative models [34, 14]. Specifically, this is done by combining two types of classifier-free guidance as in (3); one source

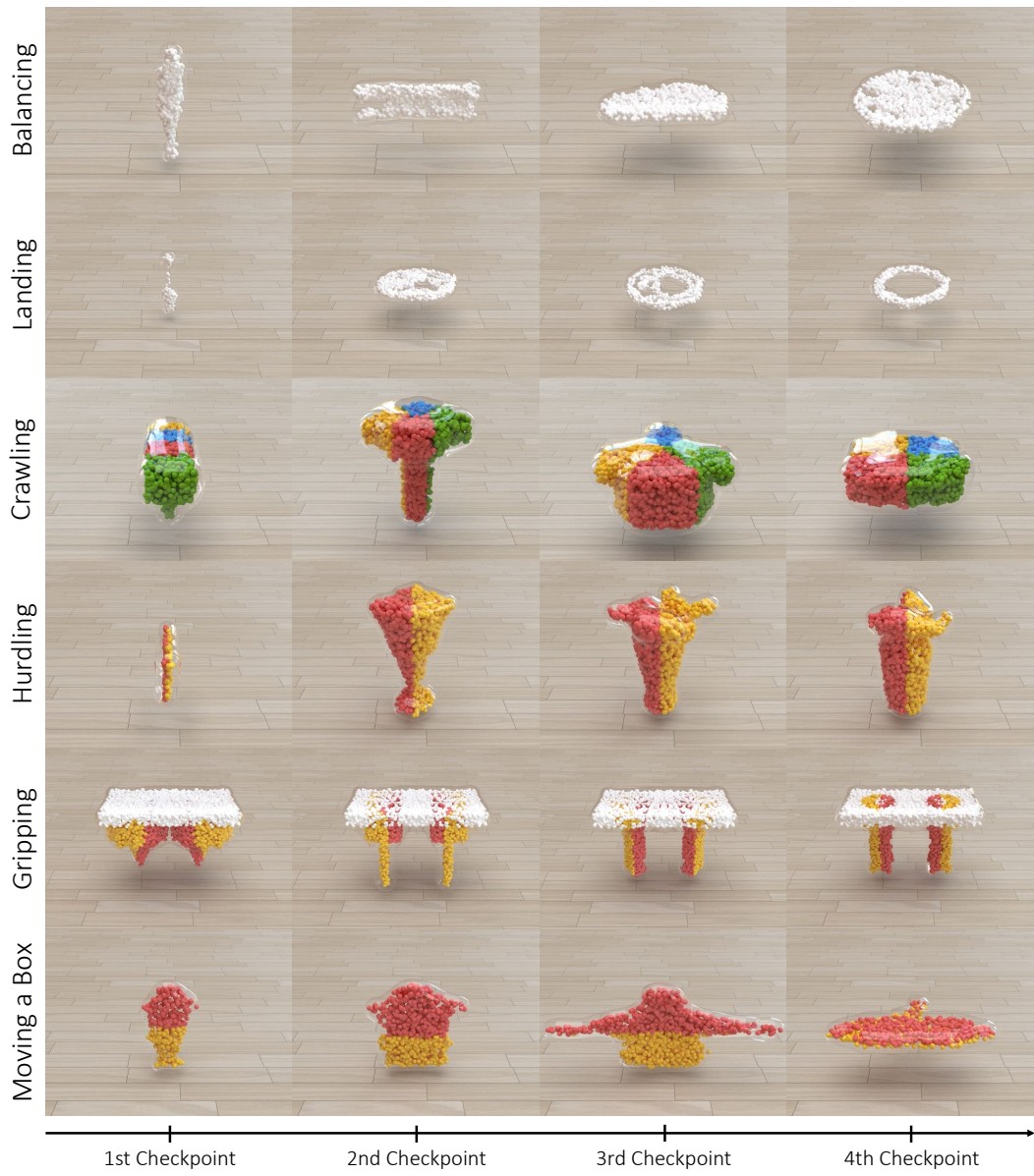

| | 1st Checkpoint | 2nd Checkpoint | 3rd Checkpoint | 4th Checkpoint |

Figure 10: Examples of DiffuseBot evolving robots to solve all presented tasks.

of guidance is derived from the task-driven embedding optimization while the other is derived from embeddings from human-provided textual descriptions. These two conditional scores, namely $\epsilon_\theta(\mathbf{x}_t, t, \mathbf{c}_{\text{physics}})$ and $\epsilon_\theta(\mathbf{x}_t, t, \mathbf{c}_{\text{human}})$, can then be integrated into the diffusion as co-design framework as in (6). We refer the reader to the original papers for more details and theoretical justification. We demonstrate examples of: the balancing robot with additional text prompt *"a ball"*, the crawling robot with additional text prompt *"a star"*, the hurdling robot with additional text prompt *"a pair of wings"*, and the moving-a-box robot with additional text prompt *"thick"*. Interestingly, human feedback is introduced as an augmented "trait" to the original robot. For example, while the hurdling robot keeps the tall, lengthy geometry that is beneficial for storing energy for jumping, a wing-like structure appears at the top of the robot body instead of overwriting the entire geometry. This allows users to design specific parts for composition while also embedding fabrication and aesthetic priorities and constraints through text. We leave future explorations of these ideas for future work.

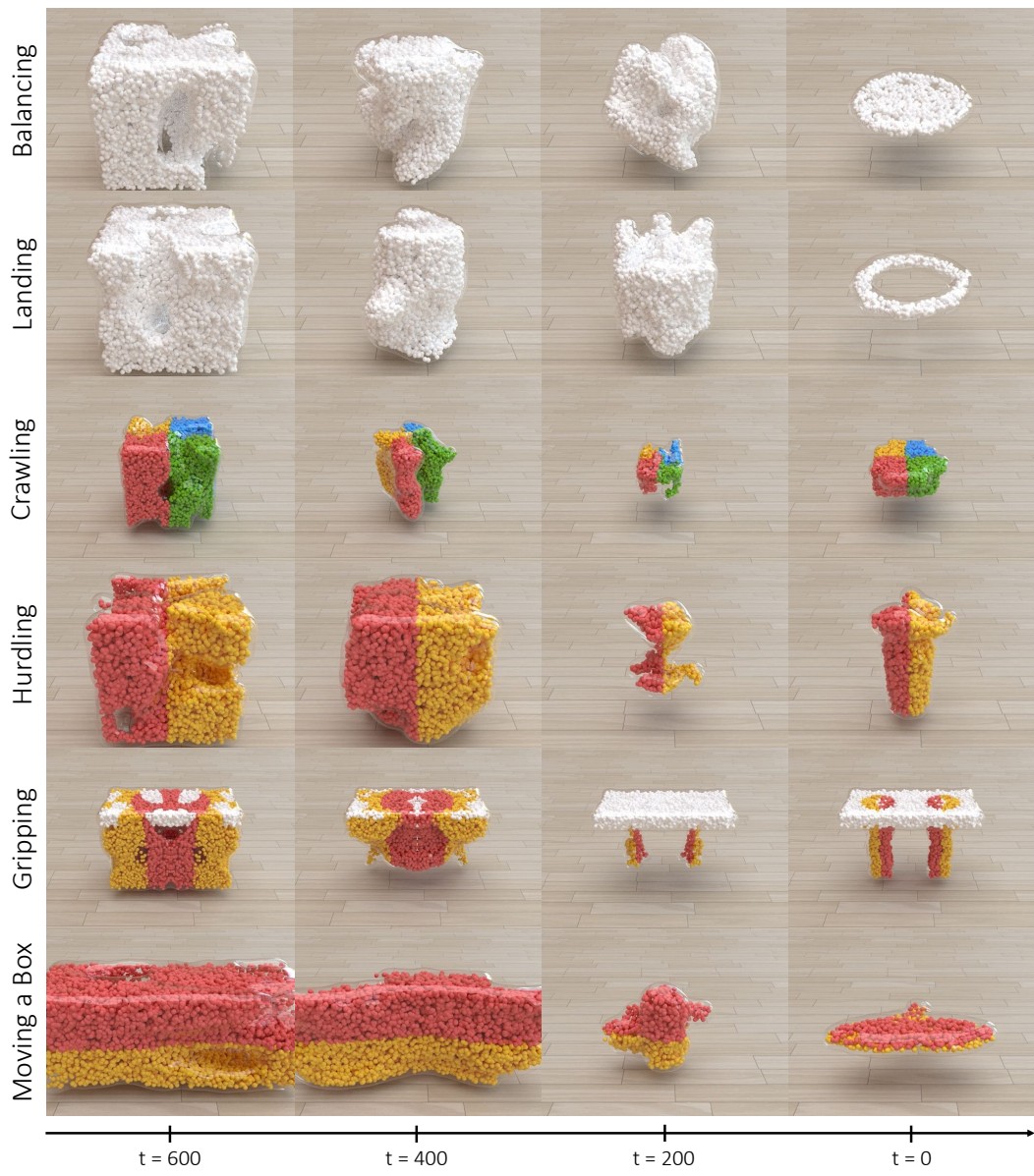

Figure 11: Demonstrations of generations changing from noises to robots with physical utility throughout the diffusion sampling process. We present snapshots at diffusion times $t = 600, 400, 200, 0$ for all presented tasks.

## G  Hardware Design and Fabrication

To create a real-world counterpart of the DiffuseBot, we fabricated a proof-of-concept physical robot gripper, as illustrated in Figure 14(a). The gripper was designed to match the shape of the digital gripper and successfully demonstrated its ability to grasp objects, as shown in Figure 15. A video demonstrating the grasping process can be found on our project page (https://diffusebot.github.io/).

During the translation from the simulated design to a physical robot, we aimed to minimize differences between the simulated and physical designs. However, due to hardware and fabrication limitations, certain modifications were necessary.

One such challenge involved replicating the arbitrary contraction force at the particles from the simulation in the physical world. To address this, we employed tendon transmission as the actuation method for the real-world gripper, which was found to be suitable for emulating interaction forces

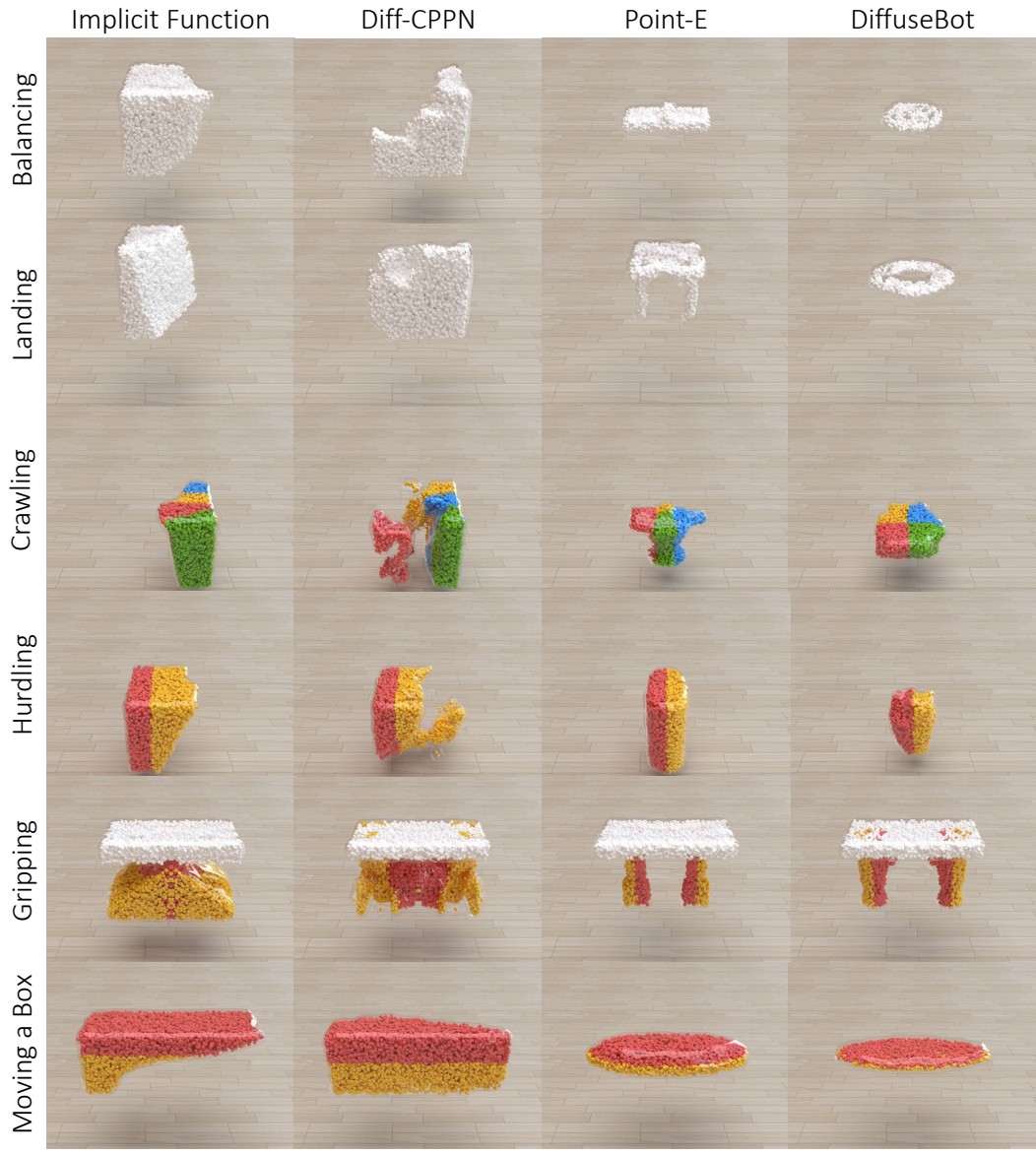

Figure 12: Comparison of robots generated by different baselines.

from the digital world without introducing significant discrepancies. To enhance the stability of the tendon-driven actuation, we incorporated a rigid base ("Base" in Figure 14(a)) above the gripper itself ("Gripper" in Figure 14(a)). This base was designed to withstand the reaction force generated by the tendon-driven actuation. Furthermore, we added four tendon routing points (represented by small dotted circles in Figure 14(b)) on each finger to securely fix the tendon path. By utilizing four Teflon tubes (shown in Figure 14(b)), we were able to position complex components such as motors, batteries, and controllers away from the gripper, reducing its complexity.

The actuation strategy of the simulated gripper applies equal contraction force to both fingers simultaneously. To replicate this strategy, we employed underactuated tendon routing in the development of the gripper. This approach eliminates the need for four separate actuators, thereby reducing the complexity of the robot. We used tendon-driven actuators specifically designed for underactuated tendon-driven robots as they solve practical issues such as size and friction issues that commonly arise in such systems [30].

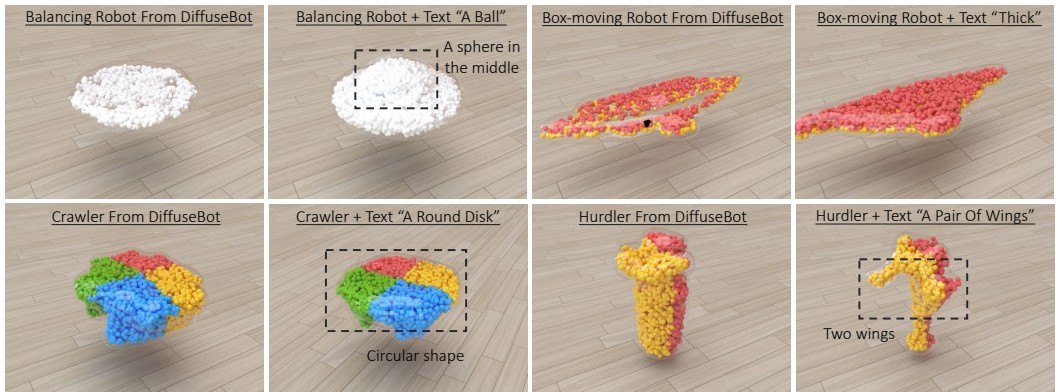

Figure 13: More examples of incorporating human textual feedback into robot generation by DiffuseBot.

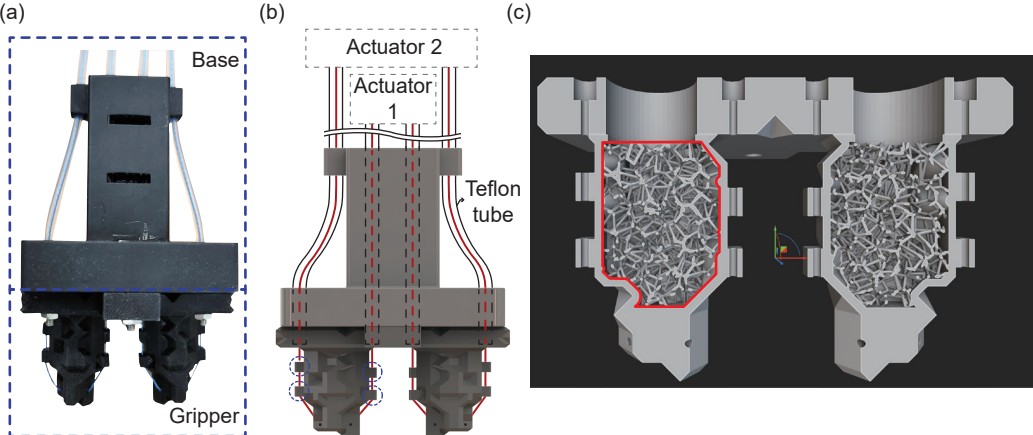

Figure 14: **(a)** shows a fabricated proof-of-concept physical robot gripper; **(b)** describes detailed robot configuration; **(c)** represents interior structure design used to soften the robot body.

The soft gripper was 3D-printed using a digital light projection (DLP) type 3D printer (Carbon M1 printer, Carbon Inc.) and commercially available elastomeric polyurethane (EPU 40, Carbon Inc.). To enhance the softness of the robot body beyond the inherent softness of the material itself, we infilled the finger body with a Voronoi lattice structure, a metamaterial useful for creating a soft structure with tunable effective isotropic stiffness [20, 36]. We generated a Voronoi lattice foam with point spacing of 2.5mm, and a beam thickness of 0.4 mm as shown in the red boundary of Fig. 14(c). Finally, we tested the soft gripper, designed and fabricated as described above, to verify its ability to grasp an object, as shown in Figure 15.

**More discussion on fabrication.** Although, at present, the compilation of the virtual robot to a physical, digitally fabricated counterpart involves manual post-processing of algorithm's output, most, if not all of these steps could be automated. Our method outputs a point cloud (defining geometry), actuator placements, and an open-loop controller, along with a prescribed stiffness. Since we can easily convert the point cloud into a 3D triangle mesh, the geometry can be created by almost any 3D printing method. In order to realize an effective stiffness and material behavior, stochastic lattices, specifically Voronoi foams, have been used [36, 20] in the past and employed here in order to match target material properties. Given the actuator placement, tendons [27, 30] can be aligned with the prescribed (contiguous) regions. Since a lattice is used, threading tendons through the robot body is simple, and we note that even more complex routings have been studied in detail in the literature [4]. Creating attachment points for the tendons is a relatively simple geometry processing problem [6]. Thus, converting a virtual robot to a specification that incorporates geometry, material, and actuation can be automated in a straightforward way.

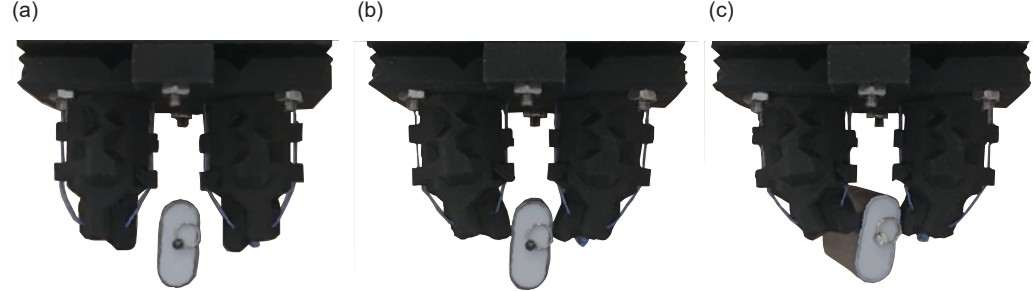

Figure 15: Grasping motion of the real-world soft gripper from DiffuseBot. Time order from **(a)** to **(c)**. The demo video can be seen at our project page.

We note that when controlled, the physical robot may not always match the virtual robot's motion. This is the sim-to-real gap, and is significantly harder to overcome in our case than translating the virtual robot to physical hardware. Significant literature has been invested in specifically tackling the sim-to-real gap, and in our case would require its own dedicated project; however, we note that often hardware can be adapted to work by modifying only the control policies using feedback from real-world experiments, often even with little human intervention [21].

**Quantitative analysis**. In Figure 16, we provide a detailed analysis on the gap between simulation and physical robot. In order to explore the quantitative gap between the behavior of the physical robot and the simulated robot, we conducted an experiment with the following conditions, where similar setups are commonly adopted in soft robot literature [17]. The objective was to measure the change in distance between two tips when we pull/release two tendons - one for closing the gripper (flexion) and the other for opening it (extension). The tendons were pulled or released in increments and decrements of 2mm.

When contracting the tendon to flex or extend the fingers, both simulation and real robot results show log-shaped graphs. The pattern in the physical robot plot is a commonly observed phenomenon called hysteresis. However, the main difference between the simulation and real-world cases can be seen when releasing the tendon from a fully contracted state. In the real robot experiment, the tip distance changes rapidly, while in the simulation, the opposite effect is observed.

One plausible explanation for this disparity could be attributed to the friction direction and elongation of the tendons. During the transition from tendon contraction to tendon release, the tension of the tendon at the end-effector may change suddenly due to the change of the friction direction. Also, since we only control the motor position (not the tendon position) to pull/release the tendon with 2mm step, the exact tendon length may not be exactly the same when we consider the tendon elongation.

**Additional real robots.** In Figure 17, we show two other fabricated robots for the moving a box task and the crawling task. For moving a box, we can achieve similar motion as in the simulation; the only slight difference occurs when the stiffness in the physical robot is not as soft as that in the simulation, falling a bit short to perform the extremely curly motion in the simulation. For crawling, we found a more significant sim-to-real gap where the physical robot cannot achieve at all the galloping-like motion in the simulation; instead, the robot crawls more alike a inchworm movement, pushing the body in a quasi-static way. The major reason is the damping of the material (elastomeric polyurethane, EPU 40, Carbon Inc.) prohibit from releasing sufficient energy for dynamical motion.

**Potential solutions**. Given that the gap between simulation and real robot performance seems to originate from the actuation/transmission method, our future work will focus on developing a tendon-driven actuation simulation framework. This framework aims to address the differences and improve the accuracy of our simulations. We are exploring other possible explanations for the sim-to-real gap and will investigate any additional factors that may contribute to the observed discrepancies. Overall, as for a high-level general solution, we believe (1) adjusting parameters based on observed sim to real gap and repeat the design process or (2) building a more accurate physics-based simulation (which can be straightforwardly plug-and-played in DiffuseBot) can largely bridge the sim-to-real gap of fabricating physical robots; or more interestingly, connecting generative models to commercial-level design and simulation softwares.

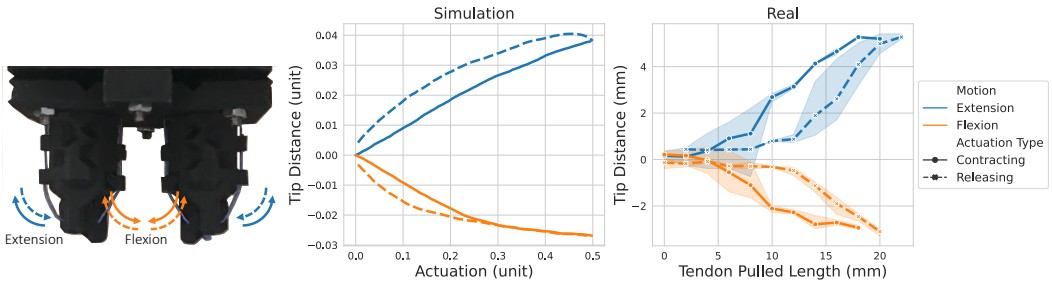

Figure 16: Quantitative analysis of behavior between simulation and physical robots.

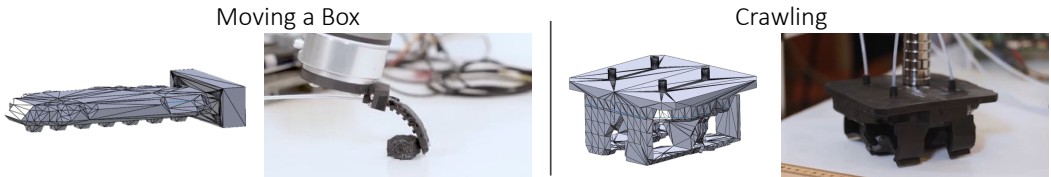

Figure 17: Other proof-of-concept real robot fabrication for moving a box and crawling.

# H    More Discussions

## H.1    Limitation

The major limitation of DiffuseBot is that we make a simplification in the parameterization of actuators and stiffness; we make dependencies of the two design specifications on robot geometry (check more technical details in Section 2.3 paragraph *Actuators and Stiffness*. This works well with properly-crafted mapping from geometry to the others yet limits the potential by human prior with little use of the generative power. While this may be reasonable as properly-set actuators and stiffness based on geometry (hand-tuned empirically in this work) roughly reflects task performance, a more flexible parameterization can definitely lead to improved performance. Potential remedy can be using part-based 3D generative models for actuators and direct optimization for stiffness. Another limitation is the gap between simulated results and real robots. While the hardware experiment has shown as a proof-of-concept that minimally demonstrates the potential, physical robot fabrication and real-world transfer have countless non-trivial challenges including stiffness and actuator design, sim-to-real gap, etc. This may require studies on more high-fidelity physics-based simulation, which can be straightforwardly plugged into DiffuseBot.

## H.2    Conditioning Beyond Text

The use of textual inputs additional to the embeddings optimized toward physical utility is achieved by both being able to be consumed by the diffusion model to produce guidance for the diffusion process . More concretely speaking, in DiffuseBot, we use the CLIP feature extractor as in Point-E and it allows to extract embedding for both text and image modalities, which can then be used as a condition in the diffusion model. Thus, we can also incorporate images as inputs and perform the exact same procedure as that of the textual inputs. Theoretically, the textual inputs are incorporated via following the intuition in end of the paragraph *Diffusion as Co-design* in Section 2.4, where the textual inputs additionally provide gradients toward following the textual specification. Similarly, the image inputs can also be processed to provide gradients since CLIP embeddings live in a joint space of images and languages. More interestingly, if we build DiffuseBot on models other than Point-E, which can consume embeddings for other modalities like audio as conditioning, we can then straightforwardly perform robot design generation guided by the other corresponding input formats (and meanwhile, toward physical utility). Note that this critical feature of compositionality across different sources of guidance throughout the reverse diffusion process is one of the biggest advantages of using diffusion-based models as opposed to other types of generative models.

### H.3 Connection To Text Inversion

There is some synergy between text inversion in [19] and embedding optimization in DiffuseBot. Both of them aim at tuning the embedding toward reflecting certain properties of the output generation, i.e., describing the output generated images in [19] and toward improved physical utility in DiffuseBot. The major difference lies in the nuance of the data/samples used to carry out the optimization. Text inversion performs a direct optimization using latent diffusion model loss (Eq. (2) in [19]), which computes losses on noisy samples/latents corrupted from the real dataset. On the other hand, it is tricky to think about real dataset in robot design (as discussed in the second paragraph of Section 1 and the paragraph *Embedding Optimization* in Section 2.4), embedding optimization in DiffuseBot computes losses on noisy samples corrupted from self-generated data filtered by robot performance (as in Algorithm 1 and Section 2.4). Conceptually, it is more like a mixture of diffusion model training and online imitation learning like DAGGER [45].

### H.4 Connection To Diffusion Models With Physical Plausibility

A potential and interesting way to adapt DiffuseBot to other applications like motion planning or control [28, 1] is to view a generated robot as one snapshot/frame of a motion/state sequence and the physics prior can be the dynamics constraint across timesteps (e.g., robot dynamics or contact dynamics that enforce non-penetration). The physics prior can be injected similarly to diffusion as co-design that propagates the enforcement of physical plausibility of generated states from differentiable physics-based simulation to diffusion samples. For example, considering states in two consecutive timesteps, we can compute loss in the differentiable simulation to measure the violation of physical constraints regarding robot dynamics or interaction with the environment. Then, we can compute gradients with respect to either control or design variables; for gradients in control, this will essentially augment works like [28, 1] with classifier-based guidance to achieve physical plausibility; for gradients in design, this will much resemble optimizing toward the motion sequence of a shape-shifting robot.

### H.5 Parameterization Of Actuator And Stiffness

The goal of DiffuseBot is to demonstrate the potential of using diffusion models to generate soft robot design and to leverage the knowledge of the pre-trained generative models learned from a large-scale 3D dataset. Under this setting, the generated output of the diffusion model can only provide the geometry information of robot designs, leading to our design choice of having full dependency of actuator and stiffness on the geometry. This may be a reasonable simplification as prior works [48] have shown geometry along with properly-set actuator and stiffness (we take manual efforts to design proper mapping from geometry to actuator and stiffness in this work) roughly reflect the performance of a soft robot design. For better generality, one potential remedy is to optimize actuator and stiffness independently from the geometry generated by the diffusion model, i.e., apply DiffuseBot and do direct optimization on actuator and stiffness afterward or at the same time. Another interesting direction may be, for actuators, to leverage part-based models [29] to decompose a holistic geometry into parts (or different actuator regions in soft robots).

