# OpenReview forum: "DiffuseBot: Breeding Soft Robots With Physics-Augmented Generative Diffusion Models"
_NeurIPS.cc/2023/Conference — NeurIPS 2023 oral_

### Official Review · Reviewer_vV19 · 2023-07-06

**Soundness:** 3 good
**Presentation:** 3 good
**Contribution:** 2 fair
**Rating:** 6
**Confidence:** 3

**Summary:**

This work proposed a new framework that augments the diffusion-based synthesis with physical dynamical simulation in order to generatively co-design task-driven soft robots in morphology and control. The extensive experiments in simulation to verify the effectiveness of DiffuseBot.

**Strengths:**

1. The paper is well-written and easy to follow.
2. The visualization is good and help me understand how the method works.
3. The application of diffusion is always be encouraged and diffusion model is an interesting and promising backbone.



**Weaknesses:**

1. The novelty is limited and more like an incremental work that applied diffusion model into 3D soft body generation task. The relationship and difference with  Softzoo mentioned in the paper should be more clearly illustrated.
2. The baseline for comparison is relatively weak, consisting of some outdated works from a few years ago. The paper should includes more strong and recent baselines.
3. The evaluation way is not agreed. 'To avoid being trapped in the local optimum, we run each baseline with 20 different random initializations and choose the best one. Since DiffuseBot is a generative method, we draw 20 samples and report the best' . If the baseline is sensitive to the initialization seed, more detailed results should be reported rather not choose the best one. And the 20 runs should report the mean and var rather not the best. The explanation about the generative model is not convincing enough for me.
4. The novelty is limited and more like an incremental work applying diffusion model in 3D generation domain.
5. The physics augmented component is not included in ablation studies and this will harm the convincing conclusion in the paper.
6. Why diffusion not other generative model, such as VAE or transformer? The ablation studies should be listed or the paper will be likely the incremental work.


**Questions:**

Please give more detailed clarification and experiments results to address my concerns.

**Limitations:**

1. The novelty is limited and the motivation of using diffusion model is not well illustrated.
2. The baseline is weak and old, the evaluation way is not agreed. The ablation studies are not enough.
3. The role and the improvement bring by physic-augmented component is not well studied.
4. The limitation part is absent in the submitted version. I hope the authors can discuss the limitations.

---

> ### Author Rebuttal · Authors · 2023-08-09
>
> We thank reviewer vV19 for bringing up several concerns. We provide additional experimental results (see the one-page pdf in the global response) and discussion as the below.
>
> **Limited novelty.**
>
> We aim to design robots with physical function and simple pattern generation will not get us there. We hence propose an entirely new framework by leveraging the computational power of a physics simulator to achieve meaningful results. Generative models are far from sufficient as:
> - The generated contents of most large-scale pre-trained generative models don’t reason about physics and fail to achieve any physical utility or functionality.
> - Unlike most training of generative modeling, computational robot design normally doesn’t have access to real data (which in our case, a dataset of well-performing robots). Instead, we need to leverage the physics-based simulation that evaluates the performance of a robot.
>
> Please refer to 1-3 paragraphs in the introduction for more detailed justification. Thus, we make the following technical contributions,
> - introduce a new framework that augments the diffusion-based synthesis with physical dynamical simulation in order to generatively co-design task-driven soft robots in morphology and control (the last paragraph in the introduction).
> - propose a method to robotize 3D shapes from diffusion samples for meaningful evaluation in physics-based simulation (section 2.3).
> - present methods for driving robot generation toward improved physical utility by optimizing input embeddings and incorporating differentiable physics into the diffusion process (section 2.4).
>
> **Stronger baselines.**
>
> As opposed to pure 3D generative modeling, soft robot co-design remains relatively unexplored. We believe most prior methods are covered as baselines for benchmarking, e.g., we adapt one of the most well-known yet a bit old method CPPN [49] to Diff-CPPN that can be used with the more recent and powerful techniques via differentiable physics. Nevertheless, we further compare with a very recent paper, DiffAqua [27]. Originally, we didn’t include this baseline since this method is designed for swimming tasks and lacks generality across a wide range of tasks. Briefly, DiffAqua proposes to compute Wasserstein barycenter among a set of primitives of underwater creatures. We report mean and standard deviation for all tasks in Table A1.
>
> We can observe that DiffuseBot outperforms DiffAqua across all tasks. There is a natural tradeoff in the method between choosing a larger set of primitives for potentially better performance across diverse tasks and obtaining good solutions in Wasserstein barycenter optimization. To this end, we believe leveraging the power of large-scale pre-trained 3D generative models remains a more scalable and general method toward soft robot co-design.
>
> **Report the mean and var.**
>
> We report the best results since normally the soft robot co-design problem expects to only produce one final robot design that can achieve high performance for a certain task (somewhat similar to other applications like drug discovery). To make the analysis more thorough, we report the mean and standard deviation in Table A2. Most results lead to conclusions that are consistent with Table 2 (using the best), except for hurdling, where the implicit function (IF) gives slightly better performance. However, IF is much more unstable, indicated as the much larger 0.63 standard deviation.
>
> **Ablation on physics components.**
>
> The ablation for the physics-augmenting components is shown in Table 1, Table 3, and Table 4 (this is the figure with caption “Varying starting...”; it is incorrectly labeled as table).
> - In Table 1, we show the results of improved physical utility by augmenting physical simulation with diffusion models; specifically, we demonstrate how the 3D diffusion model (Point-E) works poorly (1st row), and how the proposed components in DiffuseBot greatly enhance the task performance (2nd, 3rd rows). Please refer to section 3.2 paragraph “Physics-augmented diffusion” for more detailed discussion.
> - In Table 3 and Table 4, we conduct more fine-grained ablation studies on embedding optimization and diffusion as co-design. Please refer to section 3.3 for more detailed discussion.
>
> **Baselines such as VAE or transformer.**
>
> While there exist many 3D generative models other than diffusion-based models (including VAE [A1,A2], normalizing flow [A3], GAN [A4], etc.), it is non-trivial to incorporate physics priors into the generative process. Compared to other generative models, diffusion models allow us to elegantly build theoretical constructs to incorporate external knowledge like physic-based simulation as guidance throughout the iterative generative process, which is also the major technical contribution in DiffuseBot. Besides, diffusion models have emerged as the de-facto of content generation, inspiring our work to harness such power to soft robot design application.
>
> To further strengthen the paper, in Table A3, we compare with a recent VAE-based 3D generative model [A1], which outperforms other widely adopted baselines [A2-A6]. Given it is an open question to augment physics in VAE-based models (and, in fact, any other generative models), we perform direct optimization of co-design on the generated samples of [A1] to leverage physics-based simulation. With more advanced ways to inject physics prior into generative process as proposed in DiffuseBot, much superior performance can be achieved. Lastly, DiffuseBot uses a transformer-based architecture to generate 3D point clouds as in Point-E.
>
> **Limitations.**
>
> Please check the paragraph about limitations in the global response.
>
> **References**
>
> [A1] Cheng. Autoregressive 3d... ECCV 2022.
>
> [A2] Kim. Setvae: Learning hierarchical... CVPR 2021.
>
> [A3] Yang. Pointflow: 3d point... ICCV 2019.
>
> [A4] Wu. Multimodal shape... ECCV 2020.
>
> [A5] Luo. Diffusion...point cloud generation. CVPR 2021.
>
> [A6] Zhou. 3d..point-voxel diffusion. ICCV 2021.

---

> > ### Comment · Reviewer_vV19 · 2023-08-13
> > **Thanks for the rebuttal.**
> >
> > Thanks for the authors.
> >
> > Novelty is currently acceptable to me.
> >
> > I am very grateful for the efforts made by the author, especially for incorporating the VAE-based generation method (although I was anticipating seeing the advantages of diffusion over other generative models by incorporating physics constraints or knowledge as conditions in the backbone of conditioned VAE). However, considering the limited rebuttal time window, I will not be demanding more.
> >
> > Furthermore, I still believe that providing mean and variance through multiple evaluations is better than reporting only the best results. For instance, in the supplementary Table A3 provided by the author, for the locomotion task and the task of moving a box, it can be observed that the variance is particularly high. There is already significant overlap with the VAE algorithm that does not incorporate physics constraints. The author should  provide the mean and variance for the baseline in Table 2 of the main submission to enhance the persuasiveness of the results and conclusions.
> >
> > If my concerns are addressed, I will immediately increase the score.

---

> > > ### Author Response · Authors · 2023-08-13
> > > **Addressing the remaining concern**
> > >
> > > Thanks for acknowledging our effort for the rebuttal and bringing up the remaining concern about reporting the mean and variance.
> > >
> > > We will provide mean and variance for the results in Table 2 of the main submission from the supplementary Table A3 along with the discussion presented in the above to further strengthen the persuasiveness of our analysis based on your suggestion. As we cannot edit the main paper now, we will incorporate those results and changes right after we can do so.
> > >
> > > We greatly appreciate your timely follow-up on our rebuttal.

---

> > > > ### Comment · Reviewer_vV19 · 2023-08-14
> > > > **Thanks for the authors' effort**
> > > >
> > > > Considering that the author has already addressed most of my concerns, I decide to raise the score to 6. Please don't forget to include the promised part about evaluation of mean and variance in the updated version.

---

### Official Review · Reviewer_syw2 · 2023-07-08

**Soundness:** 4 excellent
**Presentation:** 4 excellent
**Contribution:** 3 good
**Rating:** 7
**Confidence:** 3

**Summary:**

This paper presents DiffuseBot, a framework that uses physics-augmented diffusion models to generate soft robot designs and control strategies for various tasks. The authors propose to optimize the embeddings conditioned by the diffusion model to improve the physical utility of the generated robots, and to reformulate the diffusion sampling process as a co-design optimization that leverages differentiable simulation. The authors demonstrate the effectiveness of their method on several tasks, such as balancing, landing, crawling, hurdling, gripping, and moving objects. They also show how to incorporate human feedback and fabricate a physical robot prototype.

**Strengths:**

The paper is well-written and clear. The proposed method is novel and interesting, combining diffusion models for shape generation, physics-based simulation and co-design optimization. The paper provides extensive experimental comparisons with baselines and ablation studies on both latent optimization and co-design to validate the proposed method. The paper also shows some fun qualitative results of diverse robot designs that function under passive dynamics, locomotion tasks and manipulation tasks.

**Weaknesses:**

- The paper does not discuss the limitations or failure scenarios of the proposed method. In addition, discussions on design choices can be help: how hyper-parameters are chosen, such as the guidance scale, or the number of MCMC steps?
- Figure 1 shows that the motivation of this work is to deploy the optimized soft robot into the real world. However, though I may miss it, I did not find discussions in the paper about the feasibility of manufacturing the resulting soft robots, such as the soft gripper. Given that the actuators are currently assumed to be muscle fibers, it can be hard for manufacturing.

**Questions:**

- Can limitations of the current pipeline be discussed and included in the paper?
- Can the paper includes discussions about manufacturing?
- How design choices are set? How sensitive is the current pipeline to hyper-parameters?
- How to interpret the metric reported in Table 1, 2? "We report the average performance with standard deviation in the superscript." -- Is the performance success rate?

**Limitations:**

Please include a section about limitations.

---

> ### Author Rebuttal · Authors · 2023-08-08
>
> We appreciate the reviewer syw2 recognizing our work as well-written, novel, and with extensive results. We address the remaining questions as the below.
>
> **Limitations or failure scenarios.**
>
> Please check the paragraph about limitations in the global response.
>
> **Hyper-parameter choices.**
>
> Hyperparameters are chosen mostly based on intuition and balancing numerical scale with very little tuning. In the following, we briefly discuss the design choices of all hyperparameters listed in Table 5 and 6 in the appendix. For min buffer size, samples per epoch, training iteration per epoch, and batch size, we roughly make sufficiently diverse the data used in the optimization and use the same setting for all tasks. For buffer size, we start with 60 and if we observe instability in optimization, we increase to 10 times, 600 (similar to online on-policy reinforcement learning); note that buffer size refers to the maximal size and increasing this won’t affect runtime. For buffer Top-K, we start with 6 and if we observe limited diversity of generation throughout the optimization (or lack of exploration), we double it. For $t_{max}$, $t_{min}$, and $\Delta t$ (we made some typos in Table 6, all 60’s should be 50’s), we roughly inspect how structured the generation in terms of achieving the desired robotic task to determine $t_{max}$ and modify $\Delta t$ accordingly to match the similar number of performing MCMC sampling (e.g., $t_{max}$/$\Delta t$: 400 / 50 $\approx$ 150 / 25). For the number of MCMC steps K, we simply set 3 for passive tasks and 5 for active tasks by intuition. For $\sigma$, we simply follow one of the settings in [11]. For the guidance scale $\kappa$ and renorm scale, we check the numerical values between $\epsilon$ and gradient from differentiable simulation and try to make them roughly in the similar magnitude, and set the same scale for all tasks for simplicity. For $\gamma$, we set 0.001 for trajectory optimization and 0.01 for parameterized controllers based on our experience of working with differentiable physics. Overall, from our empirical findings, the only hyperparameters that may be sensitive include buffer size and buffer Top-K for optimization stability and generation diversity, and guidance scales, which need to be tuned to match the numerical magnitude of other terms so as to take proper effect.
>
> We will include the above descriptions in the appendix section C.
>
> **Details on robot manufacturing.**
>
> The details of the physical robot experiment and the manufacturing is in the appendix section G. We describe how to build muscle fiber with tendon-driven actuators, how to achieve soft bodies with lattice structure, and how to fabricate the physical robot with a carbon 3D printer. Please find more details in the section G along with videos of using the soft gripper to pick up various types of objects in the project site (link shown in line 685). In addition, we further conduct a simple quantitative analysis on the behavior of simulation and the physical robot. Please check out more details in the response to reviewer 7Vvn.
>
> Although, at present, the compilation of the virtual robot to a physical, digitally fabricated counterpart involves manual post-processing of algorithm's output, most, if not all of these steps could be automated. Our method outputs a point cloud (defining geometry), actuator placements, and an open-loop controller, along with a prescribed stiffness. Since we can easily convert the point cloud into a 3D triangle mesh, the geometry can be created by almost any 3D printing method. In order to realize an effective stiffness and material behavior, stochastic lattices, specifically Voronoi foams, have been used [A1,A2] in the past and employed here in order to match target material properties.  Given the actuator placement, tendons [A3,A4] can be aligned with the prescribed (contiguous) regions. Since a lattice is used, threading tendons through the robot body is simple, and we note that even more complex routings have been studied in detail in the literature [A5]. Creating attachment points for the tendons is a relatively simple geometry processing problem [A6]. Thus, converting a virtual robot to a specification that incorporates geometry, material, and actuation can be automated in a straightforward way.
>
> We note that when controlled, the physical robot may not always match the virtual robot's motion.  This is the sim-to-real gap, and is significantly harder to overcome in our case than translating the virtual robot to physical hardware.  Significant literature has been invested in specifically tackling the sim-to-real gap, and in our case would require its own dedicated project; however, we note that often hardware can be adapted to work by modifying only the control policies using feedback from real-world experiments, often even with little human intervention [A7].
>
> **Metrics.**
>
> They are more of a soft version of success rate. The definition of the metric in Table 1 and 2 is in the appendix section D. We will add a pointer at line 182 in section 3.1 as,
>
> “We refer the reader to the appendix Section D for more detailed task descriptions and performance metrics.”
>
> **References**
>
> [A1] Martínez et al. "Procedural voronoi foams for additive manufacturing." TOG 2016.
>
> [A2] Goswami et al. 3D‐architected soft machines with topologically encoded motion. Advanced functional materials 2019.
>
> [A3] In et al. A novel slack-enabling tendon drive that improves efficiency, size, and safety in soft wearable robots. ToM 2016.
>
> [A4] Kim et al. Slider-tendon linear actuator with under-actuation and fast-connection for soft wearable robots. ToM 2021.
>
> [A5] Bern et al. "Interactive design of animated plushies." TOG 2017.
>
> [A6] Chen et al. Encore: 3D printed augmentation of everyday objects with printed-over, affixed and interlocked attachments. UIST 2015.
>
> [A7] Ha et al. Learning to Walk in the Real World with Minimal Human Effort. CoRL 2021.

---

> > ### Comment · Reviewer_syw2 · 2023-08-16
> > **nice work!**
> >
> > Thanks a lot for the rebuttal and the nice work! Authors cleared most of my concerns. Would like to keep my rating of 7 Accept.

---

### Official Review · Reviewer_zr6A · 2023-07-11

**Soundness:** 3 good
**Presentation:** 3 good
**Contribution:** 3 good
**Rating:** 6
**Confidence:** 2

**Summary:**

This paper introduces DiffuseBot, a physics-augmented diffusion model designed for generating and optimizing the morphologies and control mechanisms of soft robots. DiffuseBot aims to bridge the gap between virtually generated content and physical utility in the domain of soft robotics. Firstly, it combines the diffusion process with a physical simulation that serves as a performance certificate, thereby ensuring the feasibility and effectiveness of the generated designs. Secondly, it details a co-design procedure that simultaneously optimizes the physical design and control of the soft robots, leveraging insights from differentiable simulation. The paper validates the efficacy of this approach by presenting a variety of both simulated and physically fabricated robots, along with their diverse capabilities.

**Strengths:**

1. In general, the paper is well written, with only minor flaws. Even those unfamiliar with soft robot design will find the paper easy to comprehend.

2. Although diffusion models are expressive and powerful, their performance for tasks dealing with physical tasks often falls short. Thus, injecting a physics prior or 'physics-augmented diffusion model' is crucial. I think the method proposed in this paper is interesting and promising.

3. The evaluation is comprehensive and thoughtful. The physical robot is impressive.


**Weaknesses:**

Overall, I did not identify any major weaknesses in the paper, but here are a few points that could strengthen it:

1. While the writing is generally clear, certain sections could benefit from clearer exposition, such as:
*  The section on diffusion as co-design is not very intuitive, especially for audiences not familiar with soft robot design. Specifically, it should be clearer how gradient-based optimization benefits robot design and what exactly line 152's "synergy" means.
* It would be helpful if the authors clarify that the "condition" in this work actually refers to text.
2. The robot's actuator and stiffness seem oversimplified, having only constant stiffness. Given that the gradient of $\Psi_{act}$ is almost zero, it appears that the actuator and stiffness are solely determined by the geometry.
3. A similar idea of tuning in the embedding space is proposed in[1]. A discussion and connection to this existing work could be interesting.
4. In general, the method the paper uses to inject a physics prior into the generation process could be applicable to more general scenarios. Works like Diffuser[2] or Decision Diffuser[3] generate state sequences with diffusion models, but the generated states can sometimes be physically implausible. A deeper discussion about the potential of the method could make the paper stronger.

[1] Gal, Rinon, Yuval Alaluf, Yuval Atzmon, Or Patashnik, Amit H. Bermano, Gal Chechik and Daniel Cohen-Or. “An Image is Worth One Word: Personalizing Text-to-Image Generation using Textual Inversion.”, ICLR, 2023.

[2] Janner, Michael, Yilun Du, Joshua B. Tenenbaum and Sergey Levine. “Planning with Diffusion for Flexible Behavior Synthesis.”, ICML, 2022.

[3] Ajay, Anurag, Yilun Du, Abhi Gupta, Joshua B. Tenenbaum, T. Jaakkola and Pulkit Agrawal. “Is Conditional Generative Modeling all you need for Decision-Making?” ICLR, 2023.

**Questions:**


1. I do not fully understand how the k-means clustering is performed for actuator and stiffness generation. Specifically, what kind of feature is used for clustering?

2. In line 86, which structural biases are you referring to ?


**Limitations:**

see weakness.

---

> ### Author Rebuttal · Authors · 2023-08-08
>
> We thank reviewer zr6A for acknowledging that our approach is interesting and promising. We address remaining questions as the below.
>
> **Clearer exposition in diffusion as co-design.**
>
> Gradient-based optimization is shown to achieve more efficient and effective design search in soft robot co-design [19,25,44,48], especially with soft robots having a continuum of bodies and non-rigid contact. Thus, we aim to connect gradient-based optimization to diffusion-based generative processes in our work.
>
> The “synergy” in line 152 is between diffusion models and energy-based models, not robot co-design; it draws a connection to MCMC sampling in energy-based models and makes the update of diffusion process more “gradient-descent-like”. This allows us to elegantly formulate gradient-based optimization, which is commonly used in soft robot co-design, in the diffusion-based generative process. Please refer to [11,12,42] for more details about the theory and the appendix section D paragraph “Connection to MCMC” for more complete theoretical motivation.
>
> More precisely, the condition refers to the embedding either from text inputs, or from image inputs, or directly optimized in the Embedding Optimization stage to achieve physical utility.
>
> We will improve the exposition in section 2.4 paragraph “Diffusion as Co-design” in the revision.
>
> **Actuator and stiffness.**
>
> The goal of DiffuseBot is to demonstrate the potential of using diffusion models to generate soft robot design and to leverage the knowledge of the pre-trained generative models learned from a large-scale 3D dataset. Under this setting, the generated output of the diffusion model can only provide the geometry information of robot designs, leading to our design choice of having full dependency of actuator and stiffness on the geometry. This may be a reasonable simplification as prior works [48] have shown geometry along with properly-set actuator and stiffness (we take manual efforts to design proper mapping from geometry to actuator and stiffness in this work) roughly reflect the performance of a soft robot design. For better generality, one potential remedy is to optimize actuator and stiffness independently from the geometry generated by the diffusion model, i.e., apply DiffuseBot and do direct optimization on actuator and stiffness afterward or at the same time. Another interesting direction may be, for actuators, to leverage part-based models [A5] to decompose a holistic geometry into parts (or different actuator regions in soft robots).
>
> **The connection to [A1].**
>
> There is some synergy between text inversion in [A1] and embedding optimization in DiffuseBot. Both of them aim at tuning the embedding toward reflecting certain properties of the output generation, i.e., describing the output generated images in [A1] and toward improved physical utility in DiffuseBot. The major difference lies in the nuance of the data/samples used to carry out the optimization. Text inversion performs a direct optimization using latent diffusion model loss (Eq. (2) in [A1]), which computes losses on noisy samples/latents corrupted from the real dataset. On the other hand, it is tricky to think about real dataset in robot design (as discussed in line 40-44 and line 125-130), embedding optimization in DiffuseBot computes losses on noisy samples corrupted from self-generated data filtered by robot performance (as in Algorithm 1 and section 2.4). Conceptually, it is more like a mixture of diffusion model training and online imitation learning like DAGGER [A4].
>
> We will include this discussion in the revision.
>
> **Discussion on more general applications [A2,A3].**
>
> A potential and interesting way to adapt DiffuseBot to other applications like motion planning or control [A2,A3] is to view a generated robot as one snapshot/frame of a motion/state sequence and the physics prior can be the dynamics constraint across timesteps (e.g., robot dynamics or contact dynamics that enforce non-penetration). The physics prior can be injected similarly to diffusion as co-design that propagates the enforcement of physical plausibility of generated states from differentiable physics-based simulation to diffusion samples. For example, considering states in two consecutive timesteps, we can compute loss in the differentiable simulation to measure the violation of physical constraints regarding robot dynamics or interaction with the environment. Then, we can compute gradients with respect to either control or design variables; for gradients in control, this will essentially augment works like [A2,A3] with classifier-based guidance to achieve physical plausibility; for gradients in design, this will much resemble optimizing toward the motion sequence of a shape-shifting robot.
>
> We will include this discussion in the revision.
>
> **Features used in k-means clustering.**
>
> We use the 3D coordinates offset by the center of the geometry as the feature for k-means clustering. We will make it clearer in line 120 in the revision.
>
> **Structural biases in line 86.**
>
> The structural biases refer to the knowledge in the pre-trained 3D generative models learned from large-scale 3D datasets. The idea is, instead of searching for good robot designs from scratch, explore in the space of what a 3D generative model has learned, which provides biases toward diverse and sensible 3D structures. We will provide a clearer description in the revision.
>
> **References**
>
> [A1] Gal et al. “An Image is Worth One Word: Personalizing Text-to-Image Generation using Textual Inversion.”, ICLR, 2023.
>
> [A2] Janner et al. “Planning with Diffusion for Flexible Behavior Synthesis.”, ICML, 2022.
>
> [A3] Ajay et al. “Is Conditional Generative Modeling all you need for Decision-Making?” ICLR, 2023.
>
> [A4] Ross et al. A reduction of imitation learning and structured prediction to no-regret online learning. AISTATS 2011.
>
> [A5] Kaiser et al. A survey of simple geometric primitives detection methods for captured 3D data. CG 2019.

---

> > ### Comment · Reviewer_zr6A · 2023-08-14
> >
> > I thank the authors for their thorough response.
> >
> > It would be great if the authors could integrate them into the revised manuscript, particularly those changes regarding clarity.
> > Keeping the writing clear for the audience is important.
> >
> > Since I am not an expert in soft robot design, I will keep the score as is.
> >
> > Thanks!

---

### Official Review · Reviewer_7Vvn · 2023-07-19

**Soundness:** 4 excellent
**Presentation:** 4 excellent
**Contribution:** 3 good
**Rating:** 8
**Confidence:** 4

**Summary:**

The paper introduces DiffuseBot, a system that aims to simplify and automate the design of soft robots in simulation and real-world systems. DiffuseBot uses diffusion-based algorithms to co-design soft robot morphology and control for specific tasks, combining the diversity of evolutionary algorithms with the efficiency of gradient-based optimization. The system is made possible by advancements in AI-driven content generation.

However, existing generative algorithms face challenges when applied to physical soft robot co-design, such as the lack of consideration for physics and task performance. To overcome these, DiffuseBot uses physical simulation to guide the generative process of pretrained large-scale 3D diffusion models. It also develops an automatic procedure to convert raw 3D geometry into a format compatible with soft body simulation.

The system optimizes the embeddings that condition the diffusion model, skewing the sampling distribution toward better-performing robots as evaluated by a simulator. It also reformulates the sampling process to incorporate co-optimization over structure and control.

DiffuseBot has been tested on a wide range of tasks, demonstrating its superiority to comparable approaches. It also allows for human input in the robot generation process and has been used to create a proof-of-concept 3D-printed real-world robot. The paper contributes a new framework that augments the diffusion-based synthesis with differentiable physics simulation, methods for driving robot generation in a task-driven way toward improved physical utility, and extensive experiments in simulation to verify the effectiveness of DiffuseBot.


**Strengths:**

This paper is robust and comprehensive in its approach. It introduces an innovative method that applies diffusion models to the co-design of robots, representing a significant contribution to the field. The authors have ensured thorough experimental coverage by testing their system, DiffuseBot, on a diverse range of tasks. This extensive testing underscores the versatility and applicability of the proposed method. Furthermore, the paper is not limited to theoretical constructs but extends to practical, real-world applications. The authors demonstrate this by providing a proof-of-concept 3D-printed real-world robot, thereby solidifying the relevance and potential of their research in real-world scenarios.

**Weaknesses:**

The paper does not exhibit any significant shortcomings or areas of concern.

**Questions:**

In your paper and supplementary materials, you've provided a qualitative discussion on the challenges of translating simulated results into the fabrication of real robots based on the design developed in simulation. Could you delve deeper into this issue by providing more detailed, quantitative results that highlight the discrepancies between the behavior of the simulated robot and its real-world counterpart? Additionally, could you propose potential solutions aimed at minimizing this gap between simulation and reality?

**Limitations:**

The paper effectively addresses all identified limitations.

---

> ### Author Rebuttal · Authors · 2023-08-08
>
> We appreciate reviewer 7Vvn for recognizing our paper as a robust, comprehensive and innovative work supported by extensive experiments and a proof-of-concept physical robot to demonstrate the potential of future research. We address the remaining suggestions as the below.
>
> **Simulation and physical robot: quantitative analysis and potential future solution.**
>
> Thanks for bringing this extremely interesting question. To start off, we would like to highlight that the main focus of our work is to showcase the fascinating possibility of applying diffusion-based generative models to soft robot co-design by augmenting physics, and the hardware experiment is more of a proof-of-concept that minimally demonstrates the potential. The sim-to-real issue in soft robots involves materials, actuation, fabrication, and many other factors, and is an open and extremely challenging research question.
>
> In order to explore the quantitative gap between the behavior of the physical robot and the simulated robot, we conducted an experiment with the following conditions, where similar setups are commonly adopted in soft robot literature [A1]. The objective was to measure the change in distance between two tips when we pull/release two tendons - one for closing the gripper (flexion) and the other for opening it (extension). The tendons were pulled or released in increments and decrements of 2mm, and the results are depicted in Figure A1 in the one-page pdf in the global response.
>
> When contracting the tendon to flex or extend the fingers, both simulation and real robot results show log-shaped graphs. The pattern in the physical robot plot is a commonly observed phenomenon called hysteresis. However, the main difference between the simulation and real-world cases can be seen when releasing the tendon from a fully contracted state. In the real robot experiment, the tip distance changes rapidly, while in the simulation, the opposite effect is observed.
>
> One plausible explanation for this disparity could be attributed to the friction direction and elongation of the tendons. During the transition from tendon contraction to tendon release, the tension of the tendon at the end-effector may change suddenly due to the change of the friction direction. Also, since we only control the motor position (not the tendon position) to pull/release the tendon with 2mm step, the exact tendon length may not be exactly the same when we consider the tendon elongation.
>
> Given that the gap between simulation and real robot performance seems to originate from the actuation/transmission method, our future work will focus on developing a tendon-driven actuation simulation framework. This framework aims to address the differences and improve the accuracy of our simulations. We are exploring other possible explanations for the sim-to-real gap and will investigate any additional factors that may contribute to the observed discrepancies. Overall, as for a high-level general solution, we believe (1) adjusting parameters based on observed sim to real gap and repeat the design process or (2) building a more accurate physics-based simulation (which can be straightforwardly plug-and-played in DiffuseBot) can largely bridge the sim-to-real gap of fabricating physical robots; or more interestingly, connecting generative models to commercial-level design and simulation softwares.
>
> **References**
>
> [A1] Fang, B., Sun, F., Wu, L., Liu, F., Wang, X., Huang, H., Huang, W., Liu, H. and Wen, L., 2022. Multimode grasping soft gripper achieved by layer jamming structure and tendon-driven mechanism. Soft Robotics, 9(2), pp.233-249.

---

> > ### Comment · Reviewer_7Vvn · 2023-08-22
> >
> > Thank you for the detailed response. I understand that it was not the main focus of the paper, and I appreciate the detailed analysis of the potential sources of the sim2real gap. I'll keep the rating as is, and I think it's a strong and interesting work.

---

### Official Review · Reviewer_5Fqn · 2023-07-25

**Soundness:** 3 good
**Presentation:** 2 fair
**Contribution:** 3 good
**Rating:** 6
**Confidence:** 2

**Summary:**

This paper proposes a physics-augmented diffusion model that generates soft robot morphologies capable of excelling in a wide spectrum of tasks called DiffuseBot. DiffuseBot bridges the gap between virtually generated content and physical utility by (i) augmenting the diffusion process with a physical dynamical simulation which provides a certificate of performance, and ii) introducing a co-design procedure that jointly optimizes physical design and control by leveraging information about physical sensitivities from differentiable simulation. In the experiment, they showed a range of simulated and fabricated robots along with their capabilities.

**Strengths:**

1. The paper introduces a new framework that augments diffusion-based synthesis with physical dynamical simulation in order to co-design task-driven soft robots in morphology and control.
2. The method leverages optimizing input embeddings and incorporating differentiable physics into the diffusion process for driving robot generation in a task-driven way toward improved physical utility.
3. They performed experiments in simulation to verify the effectiveness of DiffuseBot, extensions to text-conditioned functional robot design, and a proof-of-concept physical robot as a real-world result.


**Weaknesses:**

The presentations in this paper were sometimes unclear, as asked in the following placeholder. The paper admitted that there are countless non-trivial challenges in the physical robot fabrication and real-world transfer, including stiffness and actuator design, and the sim-to-real gap. However, in my opinion, other contributions of this paper may outperform the weaknesses.

**Questions:**

1. In Section 2 before 2.1, there was no reference to 2.2. Is it fine?
2. I did not find the explicit definition of bold c in Eq. (5).
3. L159: what does the slash mean?
4. In the experiments, I want to know why such tasks were selected (i..e, motivation for the tasks).
5. I cannot find the explanation about the performance metric such as Tables 1 and 2 (sometimes having minus values). This information is important and should be mentioned in the main text.
6. The (short) introduction of the baseline models and the reasons can be mentioned.
7. The figure at the bottom of page 7 was Table 4, but may be incorrect. And Figure 4 in L233 may be incorrect.
8. Section 3.4: The paper discusses the use of textual inputs and is very interesting. Can the authors discuss the potential for using other input formats?


**Limitations:**

In the conclusion section, there seems to be less limited information about this work from the experimental results. In other parts, general limitations were mentioned.

---

> ### Author Rebuttal · Authors · 2023-08-08
>
> We thank reviewer 5Fqn for positive comments on the soundness and the contribution of our work. We address the remaining questions as below.
>
> **Challenges of physical robot fabrication.**
>
> Thanks for recognizing the contribution of our work in spite of these non-trivial challenges of real-world transfer. We further conduct a simple quantitative analysis on sim-to-real transfer. Please check out more details in the response to reviewer 7Vvn along with the experimental results shown in the one-page pdf in the global response as well as the in-depth discussion on manufacturing physical robots in the response to reviewer syw2.
> Note that this experiment only serves a preliminary study and more comprehensive and in-depth analysis along with further contribution should be done in the future research. We will also dedicate a paragraph to discuss these challenges and potential remedy in the limitation section in the revision.
>
> **No reference to 2.2.**
>
> We will add the following in the first paragraph of section 2 in the revision,
>
> “… then describe the proposed DiffuseBot framework, which consists of diffusion-based 3D shape generation (Section 2.2), a differentiable procedure …”
>
> **Explicit definition of \bold c.**
>
> The **c** in Eq. (5) is the embedding to be optimized. We will add a clearer definition at line 133 right after Eq. (5) in the revision.
>
> **L159: what does the slash mean?**
>
> It is a typo and it should be a period. We will update this in the revision.
>
> **Motivation for the tasks.**
>
> At a high level, we select tasks that
> 1. can cover a wide spectrum of existing robotics tasks: we briefly categorize tasks into passive dynamics, locomotion, and manipulation. Note that passive dynamics tasks are explicitly considered here since there is no active control of robot bodies, making optimization on robot design a direct factor toward physical utility.
> 2. only involve lower-level control/motion without the complications of long-term or higher-level task planning: we select tasks that mostly involve few motor skills, e.g., in manipulation, instead of pick and place, we simply aim at picking up/gripping an object.
> 3. are commonly considered in other soft robot co-design literature: all proposed active tasks are widely used in the soft robot community, including crawling [5,7,35,48], hurdling/jumping [19,A1,A2], and manipulating objects [3,8,27].
> 4. may induce more visible difference in robot designs between the performing and the non-performing ones to facilitate evaluation and algorithmic development: we select tasks more based on heuristics and intuition, e.g., in crawling, we expect leg-like structures may outperform other random designs.
>
> We will include the above discussion in the appendix section D in the revision.
>
> **Explanation about the performance metric.**
>
> The performance metrics are described in the appendix section D. We will add brief description as below and a pointer in section 3.1,
>
> “We refer the reader to the appendix Section D for more detailed task descriptions and performance metrics.”
>
> **Brief introduction to the baselines.**
>
> In the revision, we will add the following paragraph in section 3.2 with more details in the appendix:
>
> “In Table 2, we compare with extensive baselines of soft robot design representation: particle-based method has each particle possessing its own distinct parameterization of design (geometry, stiffness, actuator); similarly, voxel-based method specifies design in voxel level; implicit function uses use a shared multi-layer perceptron to map coordinates to design; DiffCPPN uses a graphical model composed of a set of activation function that takes in coordinates and outputs design specification. These baselines are commonly used in gradient-based soft robot co-design [19,44,48]. ”
>
> **Incorrect labeling of Table 4.**
>
> Thanks for catching these typos. Table 4 at the bottom of page 7 should be labeled as Figure X and the reference of Figure 4 in L233 should be Figure X. We will fix the labeling and referencing in the revision. (After fixing this issue in the manuscript, X is 6 and the original Figure 6 becomes Figure 7)
>
> **Other input formats than texts.**
>
> The use of textual inputs additional to the embeddings optimized toward physical utility is achieved by both being able to be consumed by the diffusion model to produce guidance for the diffusion process $\epsilon$. More concretely speaking, in DiffuseBot, we use the CLIP feature extractor as in Point-E and it allows to extract embedding for both text and image modalities, which can then be used as a condition $\mathbf{c}$ in the diffusion model. Thus, we can also incorporate images as inputs and perform the exact same procedure as that of the textual inputs. Theoretically, the textual inputs are incorporated via following the intuition in lines 162-165, where the textual inputs additionally provide gradients toward following the textual specification. Similarly, the image inputs can also be processed to provide gradients since CLIP embeddings live in a joint space of images and languages. More interestingly, if we build DiffuseBot on models other than Point-E, which can consume embeddings for other modalities like audio as conditioning, we can then straightforwardly perform robot design generation guided by the other corresponding input formats (and meanwhile, toward physical utility). Note that this critical feature of compositionality across different sources of guidance throughout the reverse diffusion process is one of the biggest advantages of using diffusion-based models as opposed to other types of generative models.
>
> We will include this discussion in the appendix in the revision.
>
> **Limitation in the conclusion.**
>
> Please check the paragraph about limitations in the global response.
>
> **Reference**
>
> [A1] Tolley, M.T., et al., An untethered jumping soft robot. IROS 2014.
>
> [A2] Bartlett, N.W., et al., A 3D-printed, functionally graded soft robot powered by combustion. Science 2015.

---

> > ### Comment · Reviewer_5Fqn · 2023-08-16
> > **Thank you for your response**
> >
> > Thank you for replying to my comments. I confirmed and understood them, but I am not an expert in soft robots, so I will leave my rating as is.

---

### Author Rebuttal · Authors · 2023-08-08

We thank all reviewers for their thoughtful and constructive feedback. We are encouraged to hear the reviewers acknowledge,
- that the proposed approach is robust, innovative, and extends beyond theoretical construct to practical, real-world applications (reviewer 7Vvn), interesting and promising as a crucial solution to inject a physics prior to diffusion models (reviewer zr6A), novel and interesting (reviewer syw2), and introduces a new framework that augments diffusion-based synthesis with physical dynamical simulation (reviewer 5Fqn);
- that the paper is well-written and easy to follow with overall clarity (reviewer zr6A, syw2, vV19);
- that the results verify the effectiveness of DiffuseBot (reviewer 5Fqn), are with thorough experimental coverage and extensive testing (reviewer 7Vvn), are comprehensive and thoughtful (reviewer zr6A), and provides extensive comparisons with baselines and ablation studies (reviewer syw2);
- that the proof-of-concept physical robot solidifies the relevance and potential of the research in real-world scenarios (reviewer 7Vvn), is impressive (reviewer zr6A).

In response to feedback, we provide individual responses below to address the remaining concerns from each reviewer to improve clarity of missing details and to provide additional discussion that strengthen our paper. Briefly, we summarize the added experiments and revision to the paper,
- Add quantitative analysis on the behavior of simulation and physical robots along with further discussion on physical robot fabrication.
- Add a paragraph for the discussion on limitations.
- Comparison to an additional baseline of a more recent soft robot co-design method.
- Comparison to an additional baseline of a VAE-based generative model.
- Report additional statistics including mean and standard deviation for baseline comparison.
- Add more clarification to the paper and discussion on relevant works.

For more details, please check individual responses. We thank all reviewers’ for their time and efforts! We hope our responses have persuasively addressed all remaining concerns. Please don’t hesitate to let us know of any additional comments or feedback on improvement.

Note that we include all additional experimental results in the one-page pdf submitted along with this global rebuttal response.

**A paragraph dedicated to limitations.** Re reviewer 5Fqn, syw2, vV19. We will add this to the revision.

“The major limitation of DiffuseBot is that we make a simplification in the parameterization of actuators and stiffness; we make dependencies of the two design specifications on robot geometry (check more technical details in section 2.3 paragraph Actuators and Stiffness. This works well with properly-crafted mapping from geometry to the others yet limits the potential by human prior with little use of the generative power. While this may be reasonable as properly-set actuators and stiffness based on geometry (hand-tuned empirically in this work) roughly reflects task performance, a more flexible parameterization can definitely lead to improved performance. Potential remedy can be using part-based 3D generative models for actuators and direct optimization for stiffness. Another limitation is the gap between simulated results and real robots. While the hardware experiment has shown as a proof-of-concept that minimally demonstrates the potential, physical robot fabrication and real-world transfer have countless non-trivial challenges including stiffness and actuator design, sim-to-real gap, etc. This may require studies on more high-fidelity physics-based simulation, which can be straightforwardly plugged into DiffuseBot.”

---

### Author Response · Authors · 2023-08-12
**Thank you and we are looking forward to your post-rebuttal feedback!**

Dear AC and all reviewers:

Thanks again for all the insightful comments and advice, which helped us improve the paper's quality and clarity.

The discussion phase has been on for several days and we have not heard any post-rebuttal responses yet.

We would love to convince you of the merits of the paper. Please do not hesitate to let us know if there are any additional experiments or clarification that we can offer to make the paper better. We appreciate your comments and advice.

Best,

Author

---

### Comment · Area_Chair_Hp46 · 2023-08-15
**Please read and respond to authors' rebuttals**

Dear reviewers,

Thank you for your reviews. The authors have posted their rebuttal. If you have not yet done so, please read the rebuttal and the other reviews, and comment on whether the rebuttal has addressed your comments or concerns.

---

### Decision · Program_Chairs · 2023-09-21

**Decision:**

Accept (oral)

**Comment:**

This paper presents DiffuseBot, a physics augmented diffusion model that can be used to design soft robots and the control in simulation and in the real-world. The initial reviews were mostly positive. The concerns raised in the reviews were sufficiently addressed in the rebuttal. All reviewers unanimously voted for acceptance. Please incorporate the contents in the rebuttal and reviewer's suggestions into the final version of this paper.